# Single-cell insights into immune dysregulation in rheumatoid arthritis flare versus drug-free remission

Kenneth F. Baker [1,2] ✉, David McDonald[3], Gillian Hulme[3], Rafiqul Hussain[4], Jonathan Coxhead[4], David Swan [5], Axel R. Schulz [6], Henrik E. Mei [6], Lucy MacDonald[7], Arthur G. Pratt [1,2], Andrew Filby [3], Amy E. Anderson [1] & John D. Isaacs [1,2]

Immune-mediated inflammatory diseases (IMIDs) are typically characterised by relapsing and remitting flares of inflammation. However, the unpredictability of disease flares impedes their study. Addressing this critical knowledge gap, we use the experimental medicine approach of immunomodulatory drug withdrawal in rheumatoid arthritis (RA) remission to synchronise flare processes allowing detailed characterisation. Exploratory mass cytometry analyses reveal three circulating cellular subsets heralding the onset of arthritis flare – CD45RO+PD1hi CD4+ and CD8+ T cells, and CD27+CD86+CD21- B cells – further characterised by single-cell sequencing. Distinct lymphocyte subsets including cytotoxic and exhausted CD4+ memory T cells, memory CD8+CXCR5+ T cells, and *IGHA1*+ plasma cells are primed for activation in flare patients. Regulatory memory CD4+ T cells (Treg cells) increase at flare onset, but with dysfunctional regulatory marker expression compared to drug-free remission. Significant clonal expansion is observed in T cells, but not B cells, after drug cessation; this is widespread throughout memory CD8+ T cell subsets but limited to the granzyme-expressing cytotoxic subset within CD4+ memory T cells. Based on our observations, we suggest a model of immune dysregulation for understanding RA flare, with potential for further translational research towards novel avenues for its treatment and prevention.

For many years, rheumatoid arthritis (RA) was viewed as a disease of inexorable joint inflammation and destruction, with progressive and irreversible disability. However, the advent of modern disease modifying anti-rheumatic drugs (DMARDs), combined with a realisation of the importance of early treatment and treat-to-target escalation of therapy, has enabled a remarkable revolution in RA outcomes such that disease remission is now an achievable and realistic target for the majority of newly-diagnosed patients[1,2]. With these incredible advances comes a new dilemma—how best to manage long-term DMARD therapy once remission has been achieved? Interventional clinical trials have shown that around half of patients with established RA in remission can stop conventional synthetic DMARDs (csDMARDs) and achieve drug-free remission (DFR)[3–5]. However, even patients in sustained drug-induced remission can experience disease flares, which both risk incremental joint damage and limit physical function and quality of life.

[1]Translational and Clinical Research Institute, Newcastle University, Newcastle upon Tyne, UK. [2]Musculoskeletal Unit, The Newcastle upon Tyne Hospitals NHS Foundation Trust, Newcastle upon Tyne, UK. [3]Flow Cytometry Core Facility, Newcastle University, Newcastle upon Tyne, UK. [4]Genomics Core Facility, Newcastle University, Newcastle upon Tyne, UK. [5]School of Medicine, University of Sunderland, Sunderland, UK. [6]Deutsches Rheuma-Forschungszentrum Berlin, A Leibniz Institute, Berlin, Germany. [7]School of Infection and Immunity, Glasgow University, Glasgow, UK. ✉e-mail: kenneth.baker@ncl.ac.uk

DMARDs are potent drugs with potential toxicity, require expensive and labour-intensive safety monitoring, and can be viewed as a hindrance to an otherwise normal lifestyle by patients in remission[6]. Strategies of DMARD minimisation would thus reduce drug toxicity and improve quality of life, whilst simultaneously reducing treatment costs and utilisation of healthcare resources. Reliable biomarkers of remission and flare, ideally measured within easily obtained samples such as peripheral blood, could help to inform such personalised approaches to DMARD therapy in future clinical practice. However, disease flares in RA are difficult to study, not least because of their sudden and unpredictable nature. Clinical trials of controlled DMARD cessation in consenting participants provide an experimental medicine model by which to synchronise flare processes, allowing a detailed study of the underlying immunological pathways.

Here, we show exploratory analyses of high-dimensional mass cytometry and single-cell RNA sequencing data from a clinical trial of csDMARD cessation to provide insights into the underlying cellular features that distinguish flare and DFR in patients with RA. Based on our data, we suggest a conceptual model of RA flare in terms of immune dysregulation with memory CD4$^+$ T cell, memory CD8$^+$ T cell, and B/plasma cell subsets promoting RA flare processes; and a role for CD4$^+$ Treg cells in the maintenance of DFR.

## Results

### Mass cytometry reveals an increase in specific memory T and B cell subsets at flare onset

Mass cytometry data incorporating 38 surface markers were generated for paired PBMC samples obtained from 36 patients in the BioRRA study at baseline and follow-up: 20 of whom flared following DMARD cessation (on-drug remission versus off-drug flare time point), and 16 who achieved DFR (on-drug remission versus 6 months DFR time point). Demographic and clinical details of these patients are shown in Table 1.

Thirty-one distinct clusters were identified, with excellent discrimination between major PBMC lineages (Fig. 1A and Supplementary Fig. 1). Seven clusters were differentially abundant between paired samples in flare patients, versus 4 clusters in DFR patients

(Supplementary Fig. 2, Table 2 and Supplementary Data 1). Clusters with longitudinal changes in proportional abundance common to both flare and remission patients could represent effects of DMARD cessation. Therefore, to identify flare-associated cellular clusters, we looked for those with significantly greater circulating abundance at onset of flare versus baseline in flare patients, and which showed no significant longitudinal difference in DFR patients. Three subsets met these criteria: CD3$^+$CD4$^+$CD45RO$^+$ICOS$^+$PD1$^+$CD38$^{hi}$ T cells (CD4_1 cluster), CD19$^+$CD27$^+$CD86$^+$CD21$^-$ B cells (BC_1 cluster), and a subset of CD3$^+$CD4$^-$CD8$^-$ presumed γδ T cells (GDT_3 cluster) (Fig. 1B, C, I). Furthermore, CD11c$^+$CD1c$^+$CXCR5$^+$ dendritic cells (DC_2) significantly reduced in abundance at flare onset versus baseline in flare patients, with no significant change in remission patients (Fig. 1H). In addition, CD3$^+$CD8$^+$CD45RO$^+$PD1$^{hi}$ T cells (CD8_2 and CD8_4 clusters) significantly increased in abundance following DMARD cessation in both groups, especially in flare patients (Fig. 1F, G). Furthermore, the circulating abundance of CD3$^+$CD4$^+$CD45RO$^+$ICOS$^+$FoxP3$^+$ T cells (CD4_3 cluster) increased after DMARD cessation in both flare and DFR groups (Fig. 1E). None of the clusters showed significant differences in proportional abundance at baseline prior to DMARD cessation between flare and DFR patients.

### Single-cell sequencing of circulating flare-associated memory T and B cells reveals distinct cellular clusters

Having identified specific flare-associated PBMC subsets using mass cytometry, we then sought to characterise their phenotype further by single-cell RNA sequencing of paired blood samples from 12 BioRRA study patients (8 flare, 4 DFR). Prior to sequencing, fluorescence-activated cell sorting was used to enrich for CD3$^+$CD4$^+$CD45RO$^+$PD1$^{hi}$ T cells (median(IQR) 13(9–17)% of total CD4$^+$ T cells), CD3$^+$CD8$^+$CD45RO$^+$PD1$^{hi}$ T cells (median(IQR) 15(13–18)% of total CD8$^+$ T cells), and CD19$^+$ B cells. The lower B cell frequency prohibited a more restrictive sorting of these cells, and the presumed γδ T cell subset was too rare to be sorted. Simultaneous unsupervised clustering of immune transcriptional and surface protein markers (320 genes and 34 surface markers) was performed separately for each of the three sorted cell populations as described below.

**Table 1 | Clinical characteristics of the mass cytometry and scRNAseq patient cohorts**

| Characteristic | Mass cytometry cohort (n = 36) | | scRNAseq cohort (n = 12) | |
| --- | --- | --- | --- | --- |
| | Flare | DFR | Flare | DFR |
| Number of patients | 20 | 16 | 8 | 4 |
| Female sex | 9 (45) | 10 (62) | 4 (50) | 2 (50) |
| Age in years | 69 (59–73) | 65 (52–71) | 64 (50–71) | 62 (44–78) |
| ACPA positive | 13 (65) | 8 (50) | 6 (75) | 2 (50) |
| RhF positive | 13 (65) | 7 (44) | 6 (75) | 1 (25) |
| Either ACPA or RhF positive | 16 (80) | 11 (69) | 8 (100) | 3 (75) |
| ACPA and RhF double positive | 10 (50) | 4 (25) | 4 (50) | 0 (0) |
| Years since diagnosis | 6.5 (3–12) | 6 (4–11) | 6 (4–10) | 2 (2–3) |
| Months since last change in DMARD therapy | 21 (12–30) | 42 (16–74) | 28 (16–36) | 15 (9–26) |
| Months since last glucocorticoid use | 31 (24–39) | 44 (31–56) | 34 (24–39) | 22 (12–34) |
| DAS28-CRP at enrollment | 1.30 (1.02–1.54) | 1.03 (0.98–2.01) | 1.05 (0.98–1.69) | 1.48 (1.04–2.00) |
| ACR/EULAR Boolean remission at enrollment | 11 (55) | 12 (75) | 6 (75) | 4 (100) |
| Methotrexate use at enrollment | 19 (95) | 12 (75) | 8 (100) | 3 (75) |
| Sulfasalazine use at enrollment | 6 (30) | 5 (31) | 3 (38) | 2 (50) |
| Hydroxychloroquine use at enrollment | 7 (35) | 2 (12) | 2 (25) | 0 (0) |
| DAS28-CRP at flare | 3.60 (2.62–4.12) | n/a | 3.94 (2.86–4.38) | n/a |
| Days from DMARD cessation to flare | 52 (33–85) | n/a | 45 (40–68) | n/a |

Values are stated as median (interquartile range) for continuous variables, and n (%) for binary variables.
*ACPA* anti-citrullinated peptide antibody, *ACR* American College of Rheumatology, *DAS28-CRP* disease activity score in 28 joints with C-reactive protein, *DMARD* disease-modifying anti-rheumatic drug, *EULAR* European Alliance of Associations for Rheumatology, *RhF* rheumatoid factor.

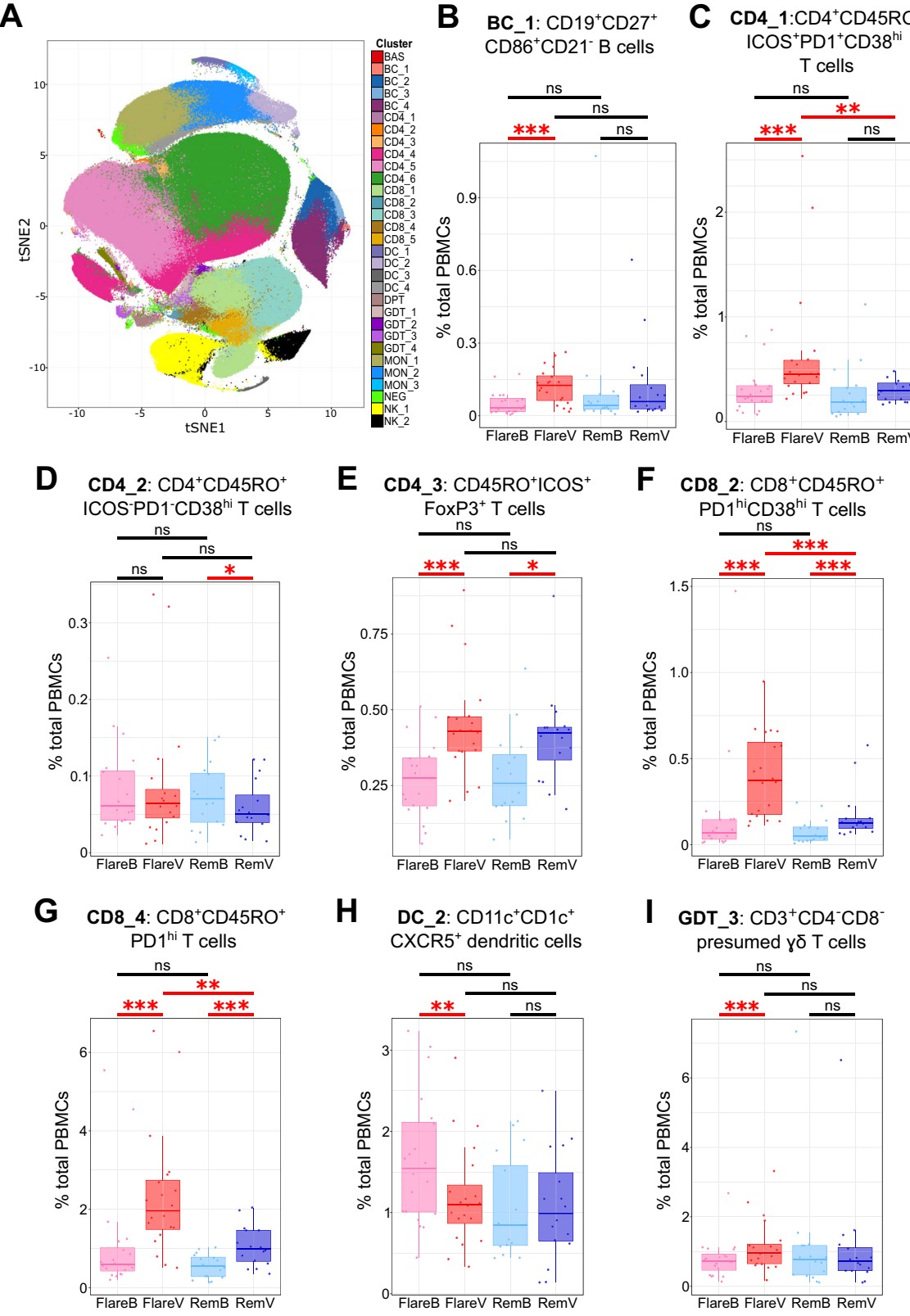

**A**

**B** BC_1: CD19$^+$CD27$^+$ CD86$^+$CD21$^-$ B cells

**C** CD4_1: CD4$^+$CD45RO$^+$ ICOS$^+$PD1$^+$CD38$^{hi}$ T cells

**D** CD4_2: CD4$^+$CD45RO$^+$ ICOS$^-$PD1$^-$CD38$^{hi}$ T cells

**E** CD4_3: CD45RO$^+$ICOS$^+$ FoxP3$^+$ T cells

**F** CD8_2: CD8$^+$CD45RO$^+$ PD1$^{hi}$CD38$^{hi}$ T cells

**G** CD8_4: CD8$^+$CD45RO$^+$ PD1$^{hi}$ T cells

**H** DC_2: CD11c$^+$CD1c$^+$ CXCR5$^+$ dendritic cells

**I** GDT_3: CD3$^+$CD4$^-$CD8$^-$ presumed γδ T cells

For CD4$^+$CD45RO$^+$PD1$^{hi}$ T cells, 17 clusters were identified with distinct profiles of surface protein and transcriptional marker expression. Of note, a small population of CD14$^+$/CD4$^{lo}$ monocytes were present within the CD4$^+$ T cell fraction (CD4_C cluster), though were clearly distinct from the other cell clusters and did not interfere with downstream analyses. For CD3$^+$CD8$^+$CD45RO$^+$PD1$^{hi}$ T cells and CD19$^+$ B cells, 12 and 11 clusters were identified respectively (Fig. 2).

**Circulating abundance and marker expression profiles of circulating cellular clusters differ between flare and drug-free remission, especially within CD4$^+$ T cells**

Next, we individually explored each cellular compartment (i.e. memory CD4$^+$ T cells, memory CD8$^+$ T cells, and B cells) to identify clusters with differential proportional abundance before and after DMARD cessation, and between flare vs. DFR patients

**Fig. 1 | Mass cytometry reveals an increase in specific memory T and B cell subsets at flare onset. A** tSNE plot of mass cytometry data showing 31 distinct circulating cellular clusters. Circulating abundance across study visits for CD19⁺CD27⁺CD86⁺CD21⁻ B cells (**B**: cluster BC_1), CD3⁺CD4⁺CD45RO⁺ICOS⁺ PD1⁺CD38ʰⁱ T cells (**C**: cluster CD4_1), CD4⁺CD45RO⁺ICOS⁻PD1⁻CD38ʰⁱ T cells (**D**: cluster CD4_2), CD45RO⁺ICOS⁺FoxP3⁺ T cells (**E**: cluster CD4_3), CD8⁺CD45RO⁺ PD1⁺CD38ʰⁱ T cells (**F**: cluster CD8_2), CD8⁺CD45RO⁺PD1ʰⁱ T cells (**G**: cluster CD8_4), CD11c⁺CD1c⁺CXCR5⁺ dendritic cells (**H**: cluster DC_2), and CD3⁺CD4⁻CD8⁻ presumed γδ T cells (**I**: cluster GDT_3). Two-sided statistical significance by generalised linear mixed model with Benjamini−Hochberg multiple test correction across all clusters within each pairwise visit comparison. ***$p < 0.001$, **$p < 0.01$, *$p < 0.05$, ns not significant. Exact adjusted $p$ values as follows: (**B**: cluster BC_1) FlareV vs. FlareB, $p = 5.9 \times 10^{-4}$; (**C**: cluster CD4_1) CD4_1: FlareV vs. FlareB: $p = 5.9 \times 10^{-4}$, FlareV vs.

RemV: $p = 5.7 \times 10^{-3}$; (**D**: cluster CD4_2) RemV vs. RemB: $p = 0.042$; (**E**: cluster CD4_3) FlareV vs. FlareB: $p = 1.7 \times 10^{-5}$, RemV vs. RemB: $p = 0.042$; (**F**: cluster CD8_2) FlareV vs. FlareB: $p = 6.3 \times 10^{-9}$, RemV vs. RemB: $p = 7.0 \times 10^{-4}$, FlareV vs. RemV: $p = 9.4 \times 10^{-4}$; (**G**: cluster CD8_4) FlareV vs. FlareB: $p = 4.4 \times 10^{-11}$, RemV vs. RemB: $p = 5.8 \times 10^{-4}$, FlareV vs. RemV: $p = 4.6 \times 10^{-3}$; (**H**: cluster DC_2) FlareV vs. FlareB: $p = 2.9 \times 10^{-3}$; (**I**: cluster GDT_3) FlareV vs. FlareB: $p = 1.2 \times 10^{-4}$. FlareB flare patient, baseline visit; FlareV flare patient, flare visit; RemB remission patient, baseline visit; RemV remission patient, month 6 visit. Box plots represent data from $n = 36$ patients (20 flare, 16 remission) where the lower bound of lower whisker shows the minimum, lower bound of box shows the lower quartile, centre of box shows the median, upper bound of box shows the upper quartile, and upper bound of upper whisker shows the maximum. Source data are provided as a Source Data file.

**Table 2 | Mass cytometry and single-cell sequencing clusters that showed significant change in proportional abundance between study visits and patient groups**

| Contrast | Cluster | Phenotype | Median % | Adj. $p$ |
|---|---|---|---|---|
| Mass cytometry ($n = 20$ flare and 16 DFR patients) | | | | |
| Flare patients: Flare onset vs. baseline (FlareV vs. FlareB) | CD4_1 | CD4⁺CD45RO⁺ICOS⁺PD1⁺CD38ʰⁱ memory T cells | 0.45 vs. 0.24 | $5.9 \times 10^{-4}$ |
| | CD8_2 | CD8⁺CD45RO⁺PD1⁺CD38ʰⁱ memory T cells | 0.37 vs. 0.07 | $6.3 \times 10^{-9}$ |
| | CD8_4 | CD8⁺CD45RO⁺PD1ʰⁱ memory T cells | 1.96 vs. 0.60 | $4.4 \times 10^{-11}$ |
| | BC_1 | CD19⁺CD27⁺CD86⁺CD21⁻ memory B cells | 0.13 vs. 0.03 | $5.9 \times 10^{-4}$ |
| | CD4_3 | CD45RO⁺ICOS⁺FoxP3⁺ memory T cells | 0.43 vs. 0.27 | $1.7 \times 10^{-5}$ |
| | GDT_3 | CD3⁺CD4⁻CD8⁻ presumed γδ T cells | 0.95 vs. 0.73 | $1.2 \times 10^{-4}$ |
| | DC_2 | CD11c⁺CD1c⁺CXCR5⁺ dendritic cells | 1.10 vs. 1.54 | $2.9 \times 10^{-3}$ |
| DFR patients: Month 6 vs. baseline (RemV vs. RemB) | CD8_2 | CD8⁺CD45RO⁺PD1⁺CD38ʰⁱ memory T cells | 0.13 vs. 0.05 | $7.0 \times 10^{-4}$ |
| | CD8_4 | CD8⁺CD45RO⁺PD1ʰⁱ memory T cells | 0.99 vs. 0.55 | $5.8 \times 10^{-4}$ |
| | CD4_2 | CD4⁺CD45RO⁺ICOS⁻PD1⁻CD38ʰⁱ memory T cells | 0.05 vs. 0.07 | 0.042 |
| | CD4_3 | CD45RO⁺PD1⁺FoxP3⁺ memory T cells | 0.42 vs. 0.26 | 0.042 |
| Flare patients baseline vs. DFR patients baseline (FlareB vs. RemB) | No significant change in cluster abundances observed | | | |
| Flare patients flare onset vs. DFR patients month 6 (FlareV vs. RemV) | CD4_1 | CD4⁺CD45RO⁺ICOS⁺PD1⁺CD38ʰⁱ memory T cells | 0.45 vs. 0.29 | $5.7 \times 10^{-3}$ |
| | CD8_2 | CD8⁺CD45RO⁺PD1⁺CD38ʰⁱ memory T cells | 0.37 vs. 0.13 | $9.4 \times 10^{-4}$ |
| | CD8_4 | CD8⁺CD45RO⁺PD1ʰⁱ memory T cells | 1.96 vs. 0.99 | $4.6 \times 10^{-3}$ |
| | BC_2 | CD19⁺CD27⁻CD86⁻CD38⁻CD21⁺ B cells | 1.20 vs. 1.96 | 0.029 |
| Single-cell RNAseq ($n = 8$ flare and 4 DFR patients) | | | | |
| Flare patients: Flare onset vs. baseline (FlareV vs. FlareB) | CD4_I | CD4⁺CD45RO⁺PD1ʰⁱCD25⁺CTLA4⁺*FOXP3+IKZF2*+ Treg cells | 1.05 vs. 0.60 | 0.044 |
| | CD4_K | CD4⁺CD45RO + PD1ʰⁱGranzyme+ T cells | 0.45 vs. 0.21 | 0.044 |
| | CD4_L | Proliferating CD4⁺CD45RO⁺PD1ʰⁱ T cells | 0.27 vs. 0.08 | 0.044 |
| | CD8_D | CD8⁺CD45RO⁺PD1ʰⁱHLA-DR⁺CD38⁺ T cells | 2.14 vs. 0.68 | 0.047 |
| | CD8_I | CD8⁺CD45RO⁺PD1ʰⁱCXCR5⁺ T cells | 0.69 vs. 0.54 | 0.047 |
| | BC_B | CD19⁺IgD⁺CD24⁺ B cells | 19.0 vs. 12.0 | 0.043 |
| | BC_G | CD19⁺CXCR3⁺ B cells | 8.13 vs. 6.55 | 0.043 |
| DFR patients: Month 6 vs. baseline (RemV vs. RemB) | No significant change in cluster abundances observed | | | |
| Flare patients baseline vs. DFR patients baseline (FlareB vs. RemB) | No significant change in cluster abundances observed | | | |
| Flare patients flare onset vs. DFR patients month 6 (FlareV vs. RemV) | No significant change in cluster abundances observed | | | |

Two-sided statistical significance by generalised linear mixed model (mass cytometry data) and Wilcoxon test (sequencing data), with Benjamini–Hochberg correction across all clusters within each visit contrast.
FlareB flare patient, baseline visit; FlareV flare patient, flare visit; RemB remission patient, baseline visit; RemV remission patient, month 6 visit.

(Supplementary Data 2). Three CD4⁺CD45RO⁺PD1ʰⁱ T cell clusters were significantly increased at flare onset versus baseline in flare patients: CD25⁺CTLA4⁺ *FOXP3+IKZF2*+ (CD4_I), granzyme + CD38⁺ (CD4_K) and proliferating (CD4_L) clusters. Furthermore, two CD8⁺CD45RO⁺PD1ʰⁱ T cell clusters were significantly increased at flare onset in flare patients: HLA-DR⁺CD38⁺ (CD8_D) and CXCR5⁺ (CD8_I) clusters. Finally, two CD19⁺ B cell subsets were also significantly increased at flare onset in flare patients: IgD⁺CD24⁺ (BC_B) and CXCR3⁺ (BC_G) clusters. Taken together, these results demonstrate an imbalance of circulating lymphocyte subsets that

is associated with arthritis flare (Fig. 3 and Table 2). In contrast, no significant differences in circulating abundance of any lymphocyte subsets were observed between flare and DFR patients at baseline (Supplementary Fig. 3).

We then compared gene and surface protein expression within each cluster between flare versus DFR patients, before and after DMARD cessation (Supplementary Data 3). Across all cell types, maximal differential marker expression was observed when contrasting flare and DFR groups compared to within-group contrasts (Fig. 4A–C).

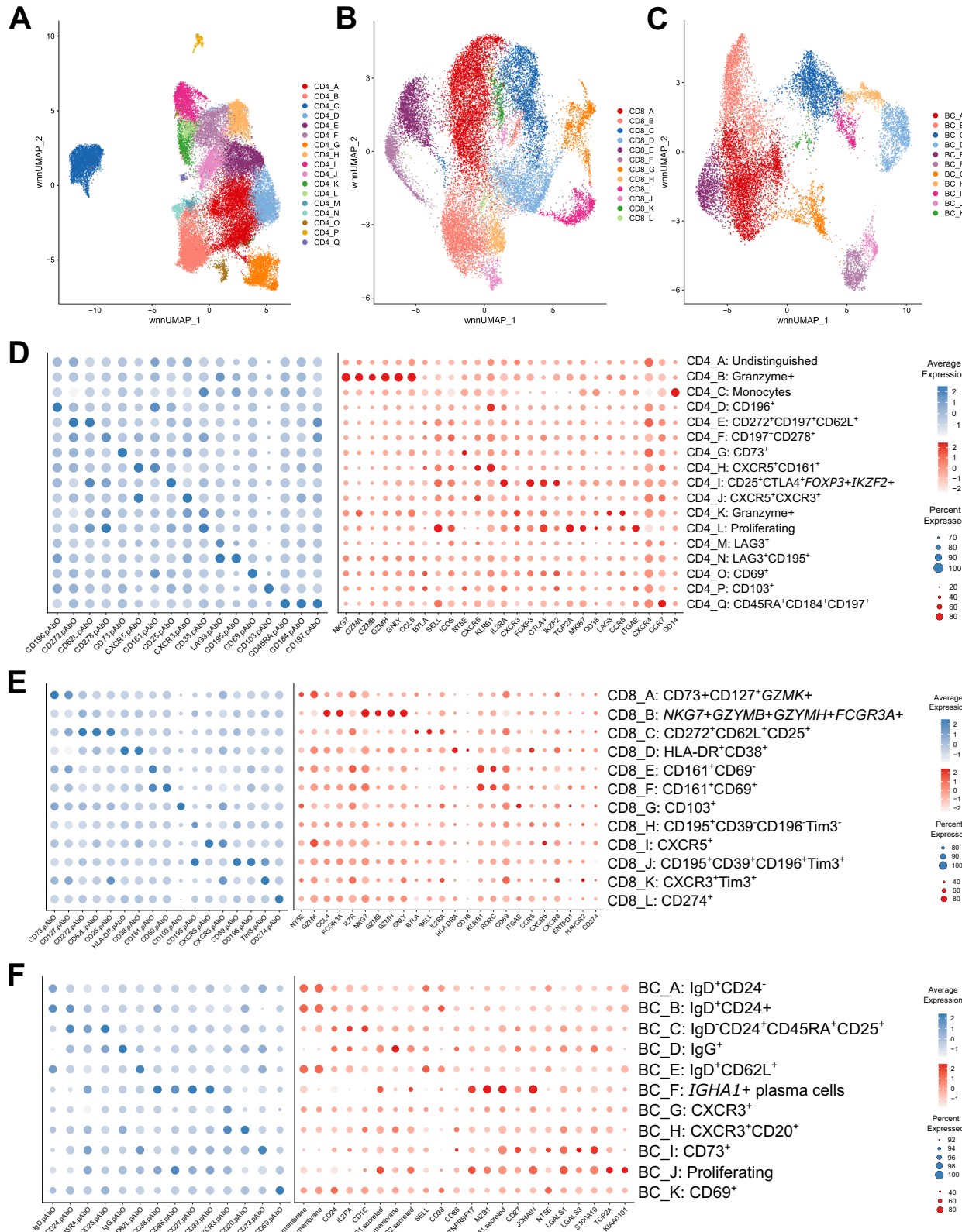

**Fig. 2 | Single-cell sequencing of circulating flare-associated memory T and B cells reveals distinct cellular clusters.** UMAP showing unsupervised clustering of CD4⁺CD45RO⁺PD1^hi T cells (**A**), CD8⁺CD45RO⁺PD1^hi T cells (**B**), and CD19⁺ B cells (**C**).

Dot plots showing average surface proteins (blue) and gene (red) expression across CD4⁺ T cell (**D**), CD8⁺ T cell (**E**), and B cell (**F**) clusters. Source data are provided as a Source Data file.

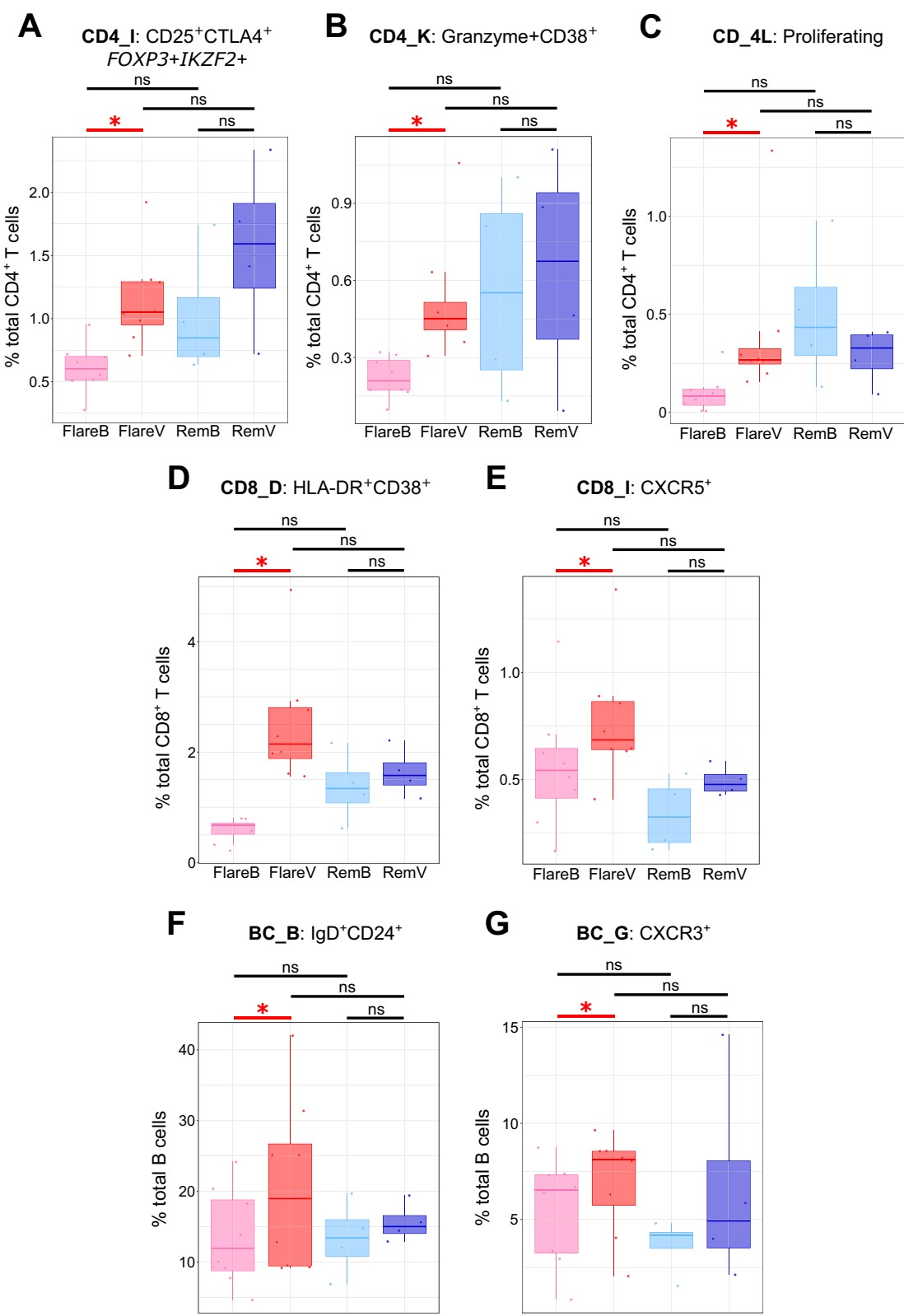

**Circulating memory CD4$^+$ regulatory T cells (Treg cells) increase at onset of flare, though with a gene expression profile distinct to DFR**

A cluster of CD4$^+$CD45RO$^+$PD1$^{hi}$ memory T cells with a robust regulatory phenotype (CD4_I cluster: CD25$^+$CTLA4$^+$ *FOXP3+IKZF2+*) was significantly increased in abundance at flare onset versus baseline in flare patients (Fig. 3A and Table 2). This cluster also showed a trend towards lower levels at baseline in flare vs. DFR patients, though this was not significant after multiple test correction. Prior to DMARD cessation, higher expression of CD274 (PDL1) and *CD7* was observed in flare versus DFR patients, which was maintained at onset of flare following DMARD cessation (Fig. 4D, H). Signalling via PDL1 on human CD4$^+$ Treg cells has been shown to have an important role in inducing a regulatory phenotype in healthy individuals by reducing ERK

**Fig. 3 | Significant differences in proportional abundance of scRNAseq clusters at onset of flare.** Circulating proportional abundance across study visits for CD4$^+$CD25$^+$CTLA4$^+$*FOXP3*+*IKZF2*+ T cells (**A:** cluster CD4_I), CD4$^+$Granzyme + CD38$^+$ T cells (**B:** cluster CD4_K), proliferating CD4$^+$ T cells (**C:** cluster CD4_L), CD8$^+$HLA-DR$^+$CD38$^+$ T cells (**D:** cluster CD8_D), CD8$^+$CXCR5$^+$ T cells (**E:** cluster CD8_I), IgD$^+$CD24$^+$ B cells (**F:** cluster BC_B) and CXCR3$^+$ B cells (**G:** cluster BC_G). Two-sided statistical significance by Wilcoxon rank sum test with Benjamini−Hochberg multiple test correction across all clusters within each cell type and pairwise visit comparison. *$p$ < 0.05, ns not significant. Exact adjusted $p$ values for FlareV vs.

FlareB contrast as follows: (**A:** cluster CD4_I) $p$ = 0.044; (**B:** cluster CD4_K) $p$ = 0.044; (**C:** cluster CD4_L) $p$ = 0.044; (**D:** cluster CD8_D) $p$ = 0.047; (**E:** cluster CD8_I) $p$ = 0.047; (**F:** cluster BC_B) $p$ = 0.043; (**G:** cluster BC_G) $p$ = 0.043. FlareB flare patient, baseline visit; FlareV flare patient, flare visit; RemB remission patient, baseline visit; RemV remission patient, month 6 visit. Box plots represent data from $n$ = 12 patients (8 flare, 4 remission) where the lower bound of lower whisker shows the minimum, lower bound of box shows the lower quartile, centre of box shows the median, upper bound of box shows the upper quartile, and upper bound of upper whisker shows the maximum. Source data are provided as a Source Data file.

phosphorylation−an effect which was attenuated in RA donors due to elevated basal phosphorylated ERK levels[7]. Furthermore, CD7 is known to be important in maintaining CD4$^+$ Treg cell homoeostasis in murine models[8], and in humans has been shown to be a marker of activated CD4$^+$ Treg cells[9] with enhanced regulatory function[10]. Supporting these observations, we observed increased expression of other genes with important roles in promoting CD4$^+$ Treg cell function at flare onset relative to DFR, including *IKZF2* (Helios) and *PRDM1* (Blimp-1), and increased expression of the cell cycle gene *MYC* (Fig. 4H). However, whereas these observations suggest increased activation of this memory CD4$^+$ Treg cell subset, paradoxical expression of other markers was observed to suggest a degree of dysfunctional regulatory capacity. Reduced expression of CD39 is seen at flare onset vs. DFR, notable as this protein plays a key role in the release of extracellular adenosine that has potent immunosuppressive effects. Indeed, reduced CD4$^+$ Treg cell expression of CD39 in patients with early RA has been shown to predict failure of response to methotrexate in RA[11,12]. Similarly, reduced expression of *FOSB* and *TRBC2* (a component of the TCR) would be in keeping with reduced cellular activation. Furthermore, increased expression of *IFITM2* (an interferon-induced protein with pro-inflammatory antiviral properties) suggests reduced regulatory capacity; for example, increased *IFITM2* expression in circulating CD4$^+$ Treg cells has been associated with checkpoint-inhibitor induced autoimmunity in patients with melanoma[13]. In summary, these observations show that despite an increase in circulating memory CD4$^+$ Treg cell abundance at onset of flare, transcriptional profiling suggests dysfunctional regulatory capacity in comparison to that observed in DFR.

## Exhausted CD4$^+$ effector T cells are primed for activation in flare patients prior to DMARD cessation

We identified a distinct cluster of CD4$^+$ T cells with an exhausted phenotype (CD4_M cluster: CD45RO$^+$PD1$^{hi}$LAG3$^+$). Whilst the circulating abundance of these cells was comparable across groups, a marked difference in gene expression was observed between flare versus DFR patients prior to DMARD cessation (Fig. 4E). Differential marker expression within this cellular subset suggests increased activation and stimulation in flare patients, including increased expression of HLA class II and *CD86*, the latter of which has been shown to be upregulated in CD4$^+$ T cells by exogenous IL2 exposure[14] and has been proposed as a marker of CD4$^+$ T cell activation[15]. Reduced expression of inhibitory and regulatory markers in flare samples including CD272 (BTLA4), CD25 and CD127 suggest an enhanced effector phenotype. However, we also observed an increased expression of Tim3, and reduced expression of CD27 and CD28, in flare cells suggesting immunosenescence[16]. Furthermore, reduced *IL32* expression was observed in flare−expression of which in CD4$^+$ T cells has been linked with activation-induced cell death[17]. This phenotype was maintained following DMARD cessation with higher expression of CD86 and HLA class II, and lower expression of CD27, CD28, CD25, CD127 and *IL32* at flare onset versus DFR (Fig. 4I). Taken together, these observations suggest chronic stimulation and exhaustion of CD4$^+$CD45RO$^+$PD1$^{hi}$LAG3$^+$ T cells in flare patients prior to DMARD cessation, which persists at flare onset.

## Cytotoxic CD4$^+$ T cells increase in abundance and expression of cytotoxic effector markers at flare onset

We identified a subset of cytotoxic CD4$^+$ T cells (CD4_K: CD4$^+$CD45RO$^+$PD1$^{hi}$granzyme + CD38$^+$) which showed distinct transcriptional profiles between flare and DFR. Prior to DMARD cessation, relatively little differential marker expression was observed, with reduced expression of *GNLY* (granulysin) and *CASP5* (caspase 5) in flare patients (Fig. 4F). In contrast, at flare onset versus DFR there was a robust upregulation of genes associated with cytotoxic T cell effector function including *NKG7*, *GZMK*, *PRF1* and *CST7* (Fig. 4J). Flare cells also showed reduced expression of CD183 (CXCR3) and CD25, in addition to reduced expression of *FOSB* and *JUNB* which have been associated with CD8$^+$ cytotoxic T cell exhaustion in chronic viral infection[18]. Together, these observations strongly suggest proliferation, activation and enhanced effector functions of cytotoxic CD4$^+$ T cells in RA flare processes.

## *IGHA1*+ plasma cells display increased maturation markers in flare versus DFR

We identified a cluster of *IGHA1*+ plasma cells (BC-F: *TNFRSF17* + *MZB1* + CD24$^-$CD38$^{hi}$) which expressed higher levels of maturation markers in flare compared to DFR patients. At baseline prior to DMARD cessation, cells in this cluster in flare patents expressed higher levels of CD196 (CCR6) and *ITGB2* (CD18) (Fig. 4G), both of which have been linked with effector functions in memory B cell lineages[19,20]. Furthermore, cells from flare patients expressed higher levels of CD274 (PDL1), which is known to be important in germinal centre B cell survival and the generation of plasma cells in animal models[21]. At onset of flare, high expression of CD274 was maintained though IgD expression was less than that seen in DFR, suggesting a potential increase in antibody-producing cells that have recently undergone class-switching (Fig. 4K).

## CXCR5$^+$ CD8 T cells are activated and increase in abundance at onset of flare

CD8$^+$CD45RO$^+$PD1$^{hi}$CXCR5$^+$ (CD8_I cluster) significantly increased in circulating abundance at onset of arthritis flare versus baseline in flare patients (Fig. 3E). Furthermore, differential marker expression analysis of this subset showed increased expression of CD38, *HLA-DPA1* and *GZMH* at flare onset relative to DFR, strongly suggesting activation of these cells during RA flare processes (Supplementary Fig. 4A). Furthermore, expression of CD183 (CXCR3) and CD127 (IL7R) were both reduced in flare versus DFR−the relevance of the latter being that reduced CD127 expression has been observed in effector versus central memory CD8$^+$ T cells[22]. Similar changes in marker expressed were observed at flare onset versus baseline in flare patients, with increased expression of HLA-DR, CD38 and *GZMH*, and reduced expression of *IL7R* (Supplementary Fig. 4B), again corroborating an increased activation and effector phenotype of these cells in RA flare.

## TCR sequencing reveals significant circulating T cell clonal expansion in flare but not in DFR patients

Having defined the surface and transcriptional phenotype of the cellular clusters, we next used single-cell CDR3 sequencing to define their

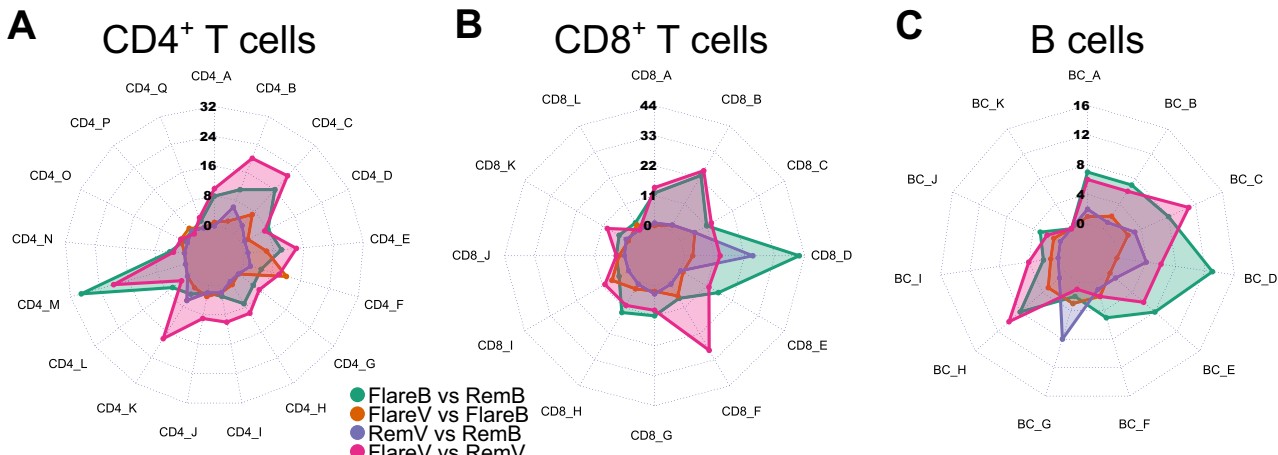

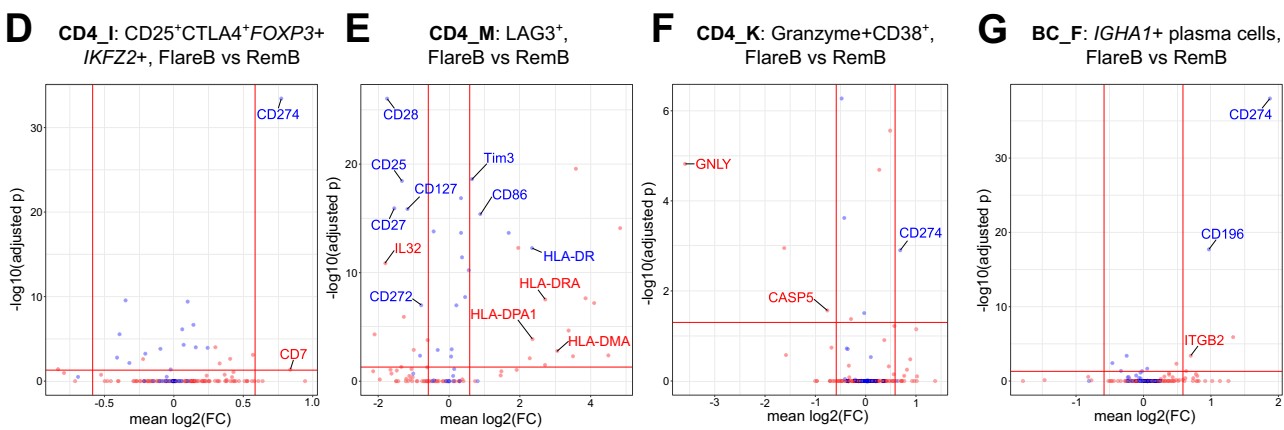

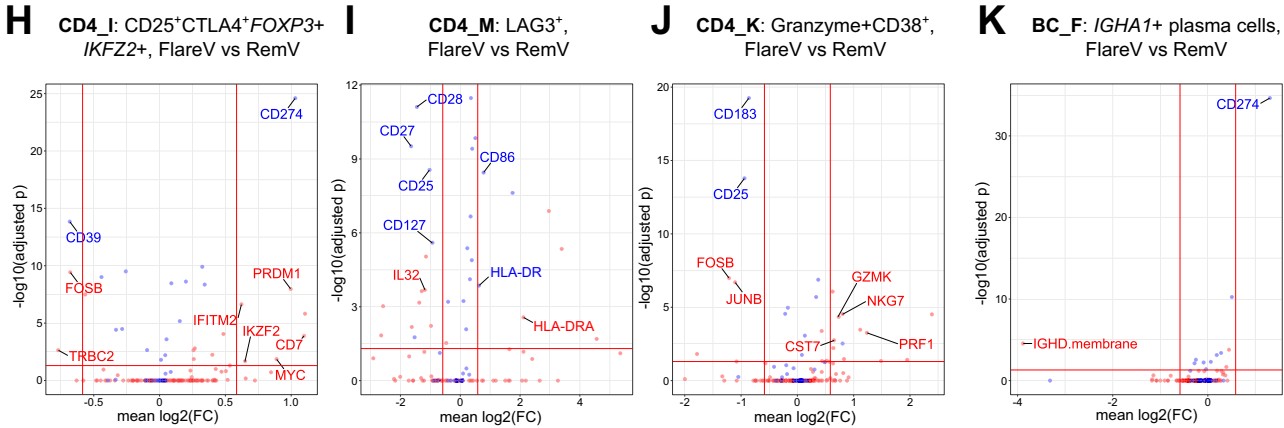

**Fig. 4 | Differential gene and surface protein expression within indicated clusters between flare and DFR.** Radar plots showing total number of differentially-expressed markers (two-sided *p* < 0.05 by Wilcoxon rank sum test with Bonferroni correction within each cluster, fold change threshold > ±1.5) within CD4⁺CD45RO⁺PD1^hi T cell (**A**), CD8⁺CD45RO⁺PD1^hi T cell (**B**), and B cell (**C**) clusters. **D**−**K** Differential gene and surface protein expression within indicated cell clusters and patient visit group contrasts. The horizontal red line indicates adjusted two-sided *p* < 0.05 threshold (Wilcoxon rank sum test with Bonferroni correction); the

vertical lines indicate ±1.5 fold change thresholds. Positive fold change values indicate an increased expression for the first group relative to second group within each contrast. Transcripts are shown in red and surface proteins in blue. Markers of interest as discussed in the results section are highlighted for reference. FlareB flare patient, baseline visit; FlareV flare patient, flare visit; RemB remission patient, baseline visit; RemV remission patient, month 6 visit. Source data are provided as a Source Data file.

clonal characteristics. A total of 17,998, 6582 and 9506 unique paired CDR3 sequences were observed for CD4$^+$CD45RO$^+$PD1$^{hi}$ T cells, CD8$^+$CD45RO$^+$PD1$^{hi}$ T cells, and B cells respectively across all patient groups (Supplementary Figs. 5–7). We defined a clone as any paired CDR3 nucleotide sequence shared by at least two cells within the same patient. Using this definition we observed 1021 unique clonal CDR3 sequences across 4618/40,783 (11.3%) CD4$^+$ T cells, 1532 unique clonal CDR3 sequences across 8850/25,520 (34.7%) CD8$^+$ T cells and 262 unique clonal CDR3 sequences across 618/15,302 (4.0%) B cells. CD4$^+$ T cell clones were disproportionately found within the CD4$^+$CD45RO$^+$PD1$^{hi}$granzyme+ subset (CD4_B), and B cell clones were disproportionately found in *IGHA1*+ plasma cells and CXCR3$^+$ B cells (BC_F and BC_G clusters). In contrast, CD8$^+$ T cell clones were widely distributed across all clusters (Fig. 5A–C and Supplementary Figs. 8–10). Significant longitudinal changes in clonal diversity (Shannon entropy) were observed before and after DMARD cessation, but with no discernible differences between flare and DFR patients (Supplementary Fig. 11).

We then searched for specific clones that significantly changed in circulating abundance after DMARD cessation within each patient. We identified 18 CD8$^+$ T cell clones (4 patients: 2 flare, 2 DFR), four CD4$^+$ T cell clones (4 patients: 2 flare, 2 DFR), and no B cell clones which showed a significant change in circulating proportional abundance at onset of arthritis flare (Supplementary Figs. 12 and 13 and Supplementary Table 1). We then analysed each subset within each patient to identify subset-specific changes in clonal abundance at an individual patient level. In CD8$^+$ T cell clusters, we observed a significant increase in subset-specific proportional abundance following DMARD cessation of 10 clones in three patients, and significant reductions of 14 clones in three patients (Fig. 5D and Supplementary Fig. 14). Within CD4$^+$ T cell clusters, we identified a significant increase in subset-specific proportional abundance following DMARD cessation of one clone in one patient, and significant reductions of 5 clones in three patients (Fig. 5E and Supplementary Fig. 15). Whereas increases in clone abundance suggest clonal expansion within the circulation, a reduction in clone abundance may signify migration of cells out of the circulation.

In summary, these data demonstrate a high level of polyclonal expansion in memory CD8$^+$ T cells following DMARD cessation, which was widespread across multiple different subsets. Clonal expansion in memory CD4$^+$ T cells was largely limited to the granzyme-expressing cytotoxic CD4$^+$ T cell subset. In contrast, no significant change in B cell clonal abundance was observed.

## Discussion

Despite a revolution in treatment outcomes over the past two decades, we understand remarkably little about what drives RA pathogenesis and what determines the balance between active disease and sustained remission. Our experimental medicine approach has allowed us to systematically study the immunological perturbations at the onset of RA flare, as well as key differences between flare and drug-free remission. Using multidimensional mass cytometry and single-cell sequencing we herein profile flare and remission in RA in detail, offering unique insights into this poorly understood immune landscape.

RA is a complex immune-mediated inflammatory disease, with dysregulation of both innate and adaptive immunity, as well as stromal cells such as synovial fibroblasts. Mass cytometry and single-cell sequencing technologies have previously been used to compare synovial tissue from patents with RA versus osteoarthritis, with enrichment of pro-inflammatory cells observed in RA synovia across a broad spectrum of cell types including CD4$^+$*PDCD1*+ T peripheral helper cells, CD8$^+$granzyme+ T cells, *ITGAX+TBX21*+ B cells, *IL1B*+ monocytes, and *THY1 + HLA-DRA*hi sublining fibroblasts[23]. A similar study has also compared synovial tissue-resident memory T cells in patients with rheumatoid arthritis and psoriatic arthritis[24]. However,

such studies lack paired longitudinal observations comparing flare and remission within the same individual, a key strength of our experimental medicine design.

One of the key findings of our study is our observation of an increased abundance of CD4$^+$ Treg cells at flare onset, though coupled with transcriptional profiles suggesting dysfunctional suppressive functions. Prior to DMARD cessation, CD4$^+$ Treg cells from flare patients showed higher expression of PDL1 and *CD7*, both markers associated with enhanced Treg cell function. However, whilst the expression of these and other regulatory markers (e.g. *IKZF2* and *PRDM1*) were higher in flare patients, the expression of other key regulatory effector markers such as CD39 was reduced. Furthermore, the expression of pro-inflammatory effector markers such as *IFITM2* were increased at flare onset. Taken together, our observations suggest a potential and biologically plausible mechanism by which dysfunctional Treg cells may predispose towards disease flare in susceptible individuals. Indeed, a potential role for CD4$^+$ Treg cells as a biomarker of DFR has recently been explored in a small ($n = 9$) pilot study, where both increased circulating Treg cell abundance and ex vivo suppressor function measured prior to csDMARD tapering was predictive of future DFR[25]. This is further corroborated by reports of increased circulating CD4$^+$ Treg cell abundance in patients who respond (versus those who are unresponsive) to methotrexate, with an inverse correlation with disease activity[26]. Similar observations exist in other IMIDs, including accumulation of clonally expanded CD4$^+$CD45RO$^+$PD1$^{hi}$ peripheral helper T (Tph) cells in synovial fluid of patients with active juvenile idiopathic arthritis[27] and evidence of dysregulated CD4$^+$ Treg cell function in type I diabetes mellitus[28]. Our results further support the potential role of Treg cells as a biomarker of tolerance in RA, and merit further study.

A further observation of our study was that *IGHA1*+ plasma cells showed significantly increased expression of maturation markers in flare versus DFR. B cells are important players in RA pathogenesis, and can contribute to immune dysregulation through antigen presentation, autoantibody production, and inflammatory cytokine production[29]. B cells likely also play a role in perpetuation of chronic synovitis–for example, expansion of circulating CD21lo/negative autoreactive B cells and plasmablasts have been observed in patients with established RA[30,31]. Furthermore, clonally-expanded autoreactive plasma cells are present in inflamed RA synovium[32], and can be driven within the synovial microenvironment by CD4$^+$ peripheral helper T cells (Tph)[33]. Approximately 70% of RA patients are seropositive for rheumatoid factor and/or anti-citrullinated peptide autoantibodies (ACPA), a proportion which rises still further when considering additional anti-modified protein antibodies (AMPAs) not routinely measured in clinical practice, such as anti-carbamylated and anti-acetylated protein antibodies[34]. ACPA positivity has been shown to be a positive predictor of flare following DMARD tapering and cessation across numerous observational and interventional studies[35]. In the RETRO study of randomised DMARD withdrawal in RA remission, positivity over a panel of 10 AMPAs showed a dose-dependent increase in the risk of flare[36]. Furthermore, baseline IgA2 levels correlated with disease activity (DAS28) at flare, whereas IgA2 ACPA levels steadily declined in those patients who maintained remission[37]. In another study, a potential pathogenic role of IgA is further supported by the enrichment within RA synovium of autoreactive B cells positive for FcRL4, an IgA receptor[38]. Furthermore, in a study of seropositive individuals at-risk for developing RA, circulating IgA$^+$ plasmablasts were elevated in comparison with individuals with early RA and healthy controls, suggesting a role of these cells in very early stages of RA pathogenesis[39]. Our findings are in keeping with these previous observations, and furthermore implicates IgA$^+$ plasma cells in the immunopathology of flare in established RA. The origin of these IgA$^+$ plasma cells cannot be ascertained in our study, but the strong link with mucosal immunity raises the intriguing possibility of host-

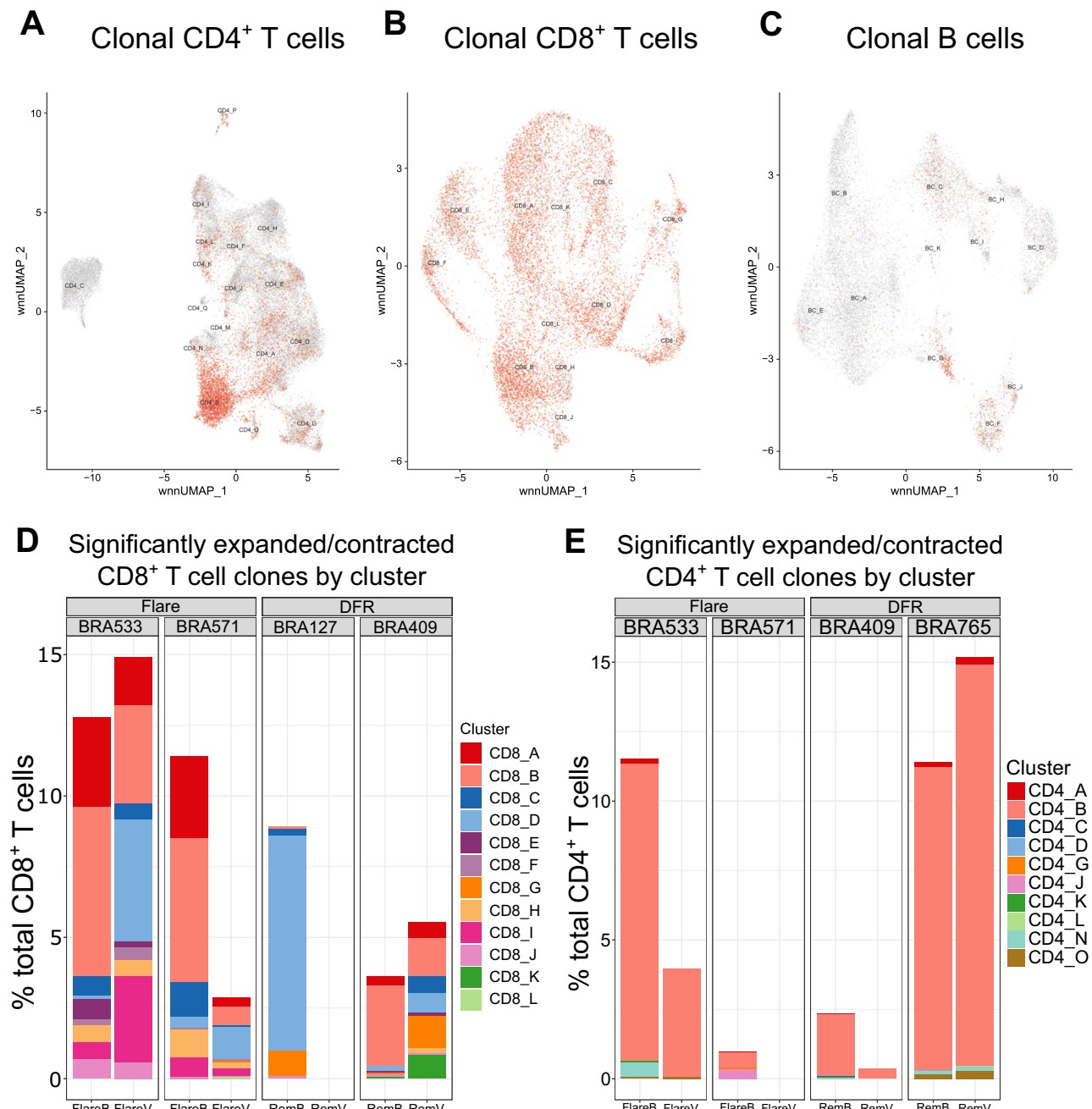

**Fig. 5 | V(D)J sequencing reveals significant T cell clonal expansion following DMARD cessation.** UMAPs showing the distribution of clonal cells (highlighted in red) within CD4+ T cells (**A**), CD8+ T cells (**B**), and B cells (**C**) including cells from all patients at all time points. Clonal cells were defined as at least two cells that shared the same paired CDR3 nucleotide sequence within the same patient. Cluster distribution of CD8+CD45RO+PD1hi T cell (**D**) and CD4+CD45RO+PD1hi T cell (**E**) clones that showed significant (two-sided $p < 0.05$, Fisher exact test with Benjamini−Hochberg correction) change in proportional abundance between visits. Each panel represents an individual patient. FlareB flare patient, baseline visit; FlareV flare patient, flare visit; RemB remission patient, baseline visit; RemV remission patient, month 6 visit. Source data are provided as a Source Data file.

microbe interactions (e.g. gut, lung etc.) as potential determinants of RA flare risk. It has previously been demonstrated that IgA from patients with established RA are reactive against gut commensal bacteria[40]. Furthermore, In K/BxN mice gut commensal segmented filamentous bacteria are required to incite RA-like inflammatory arthritis[41], with changes in IgA repertoire associated with arthritis activity[42]

Our mass cytometry analyses reveal an increased circulating abundance of CD45RO+ memory CD4+ and CD8+ T cells at onset of flare versus drug-induced remission. Both of these cell subsets show a remarkably similar phenotype, with high expression both of activation markers (CD38, Ki67, ICOS) and markers associated with immune exhaustion (PD1, CTLA-4). CD4+ T cells are known to play an important role in RA pathogenesis, not least demonstrated by the strong association of seropositive disease with HLA-DRB1 "shared epitope" alleles and its interaction with cigarette smoking[43], and multiple GWAS hits linking RA susceptibility to genes associated with CD4+ T cell function[44]. Furthermore, data exist from several studies linking activated and/or exhausted CD4+ T cell circulating frequency with disease activity in RA[45–47]. Our data further support these observations, and suggest that CD4+CD45RO+PD1hiLAG3+ exhausted T cells are primed for activation and may thus act as a crucial trigger for flare initiation.

There are relatively few studies examining the potential role of CD8+ T cells in RA pathogenesis, though emerging data suggest that they are also likely to play an important role in driving active synovitis. For example, tissue resident memory CD8+CD45RO+CD69+CD103+ T cells have been shown to persist in previously inflamed joints in both human disease and animal models, and in the latter can be activated in an antigen-specific manner to mediate recruitment of circulating effector cells and trigger arthritis flare[48]. Further evidence supports a role for CMV and EBV virus-specific CD8+ T cells in driving RA synovial inflammation[49]. CD8+PD1+CXCR5+ T cells have also been shown to support autoantibody production in animal models[50], and CD8+CD40L+ T cells were required for the formation of synovial ectopic germinal centres in a chimeric RA mouse model[51]. Our data further support a role for CD8+ T cells in RA flare, with CD8+CD45RO+PD1hiCXCR5+ T cells demonstrating enhanced expression of proliferation and activation markers at flare onset.

Our V(D)J sequencing data demonstrate widespread clonal expansion across multiple memory CD8+ T cell subsets, with further expansion within the circulation at flare onset. This mirrors observations in juvenile idiopathic arthritis, where substantial clonal expansion was observed in CD8+PD1+ T cells within synovial fluid[52]. In contrast, we observed clonal expansion in memory CD4+ T cells that was largely restricted to a cytotoxic CD4+ T cell subset expressing granzyme genes. Cytotoxic CD4+ T cells are observed in chronic viral infections such as CMV[53], are a hallmark of immunosenescence in elderly individuals[54], and accumulate in immune-mediated inflammatory diseases as a result of chronic antigen-driven expansion[55]. Indeed, cytotoxic CD4+ T cells are enriched in the circulation and synovial fluid and tissue of patients with RA[56]. Accordingly, our data demonstrate abundant clonal expansion within circulating cytotoxic CD4+ T cells, with an increase in circulating clone abundance coincident with the onset of arthritis flare. Furthermore, cytotoxic CD4+ T cells from flare patients showed a transcriptional profile that was primed for activation and tissue migration. Whilst the presence of these cells within the inflamed synovium and their antigen specificity remain to be established, our data do suggest a potential antigen-driven clonal expansion of cytotoxic CD4+ T cells as part of RA flare processes and deserves further study.

Based on our combined observations, we propose a model to describe the features that predispose towards and potentially trigger RA flare (Fig. 6). Prior to DMARD cessation, various effector memory lymphocyte subsets are primed for activation and proliferation, though this is tempered by the immunomodulatory effects of DMARD therapy and CD4+ Treg cells. Upon DMARD cessation effector memory lymphocytes are unleashed, becoming activated and upregulating pro-inflammatory effector molecules. In contrast, although circulating CD4+ Treg cell abundance increases with flare, a dysfunctional transcriptional profile suggests a reduced regulatory capacity that is insufficient to counteract the activation of pro-inflammatory effector memory lymphocytes. The net effect is thus to disrupt immune homoeostasis, triggering an arthritis flare. Our model is limited to observations of circulating lymphocytes—further work is required to elucidate the concurrent immunological perturbations within the synovial compartment.

Our experimental medicine approach to synchronise flare, combined with the richness of our single-cell datasets, offer detailed insights into the processes underpinning flare and DFR in RA. Nevertheless, our study does have some limitations. To obtain sufficient cells for analysis, our single-cell T cell sequencing was restricted to memory CD45RO+PD1hi cells as identified by our mass cytometry analyses, and thus relevant findings in other cell subsets may have been overlooked. This would include, for example, CD4+CD45RA−TNFα+PD1−CD152− cells that have been associated with disease relapse following anti-TNF withdrawal in patients with juvenile idiopathic arthritis[57]. The small size of our single-cell sequencing cohort prohibits multivariate analysis across cellular and clinical variables. Furthermore, we advise caution when comparing the magnitude and statistical significance of longitudinal changes in proportional abundance and marker expression between flare and remission patients, owing to fewer remission patients and thus less statistical power in this group. The longitudinal nature of our study, combined with the need to minimise variability in sample processing, necessitated the use of frozen cells and thus markers not robust to cryopreservation may have been depleted prior to analysis. Our study excluded patients taking biologic DMARDs—whether our findings are consistent in those taking these potent and specific immunomodulatory drugs remains to be established.

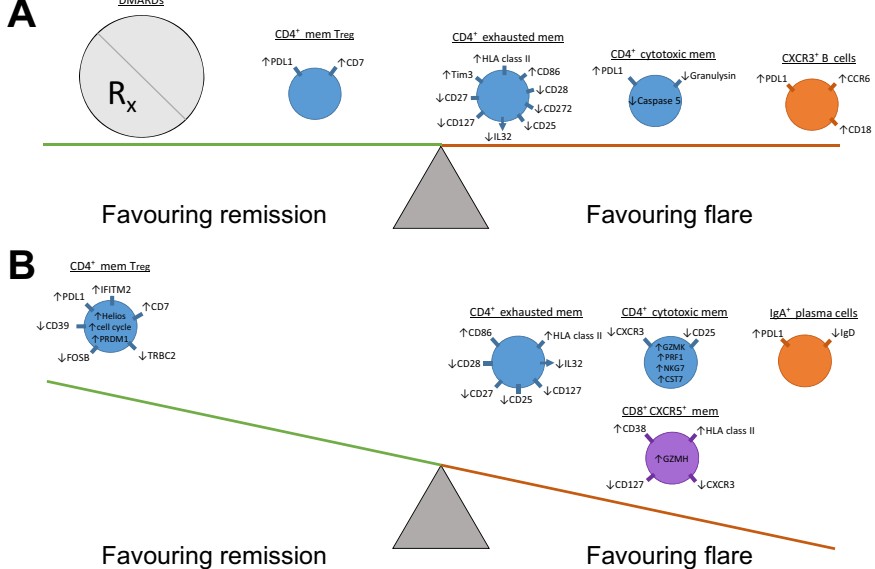

**Fig. 6 | Model describing the changes in circulating lymphocyte subsets that predispose towards and potentially trigger RA flare. A** In drug-induced remission, effector lymphocyte subsets are primed for activation and proliferation, but are balanced by the immunomodulatory effect of DMARDs, and attenuated CD4+ Treg cell suppression. **B** Upon DMARD cessation, pro-inflammatory effector lymphocyte subsets are activated, overwhelming a dysfunctional CD4+ Treg cell response. Mem memory.

Furthermore, patient outcome data is limited by the relatively short 6-month follow-up duration in BioRRA study. Given this, the relevance of our findings in relation to sustained DFR beyond 6 months (i.e. true immunological remission versus transient disease control) is uncertain, though ongoing and future work to ascertain long-term outcomes in our cohorts will illuminate this further. Similarly, future work to explore the synovial microenvironment during RA flare will further enrich and build upon our findings. Our data suggest potential immune cell subsets that may play a role in flare and DFR processes and thus may hold potential value as clinical biomarkers, though we cannot generalise our findings outside of the BioRRA cohort based on this dataset alone. Further work would be required to validate our observations in external cohorts, including both healthy and disease controls, before our findings could be translated towards future clinical practice. Finally, our study provides exploratory observational rather than empirical functional data–all mechanistic associations are therefore by theoretical inference only.

In conclusion, our data provide crucial insights towards understanding the enigmatic cellular immune mechanisms underpinning the genesis of flare and maintenance of remission in RA. Our observations implicate specific effector lymphocyte subsets in promoting RA flare and CD4$^+$ Treg cells in maintaining drug-free remission. Our model of RA flare suggests potential avenues for future translational research which, if validated, could inform the development of future strategies to predict and prevent RA flare. Clearly much work remains to be done to achieve such a paradigm shift towards personalised therapy in RA. However, if successful it would herald a second therapeutic revolution in the management of this common IMID, with widespread benefits for patients, clinicians and society.

## Methods

### Study design

Exploratory mass cytometry and single-cell RNA sequencing analyses were performed using peripheral blood mononuclear cell (PBMC) samples from the Biomarkers of Remission in Rheumatoid Arthritis (BioRRA) study[5]. Patients with established RA in remission (DAS28-CRP ≤ 2.4 and absence of power Doppler synovitis on 7-joint ultrasound) taking csDMARDs (methotrexate, sulfasalazine and/or hydroxychloroquine), and without current or recent use of biologics (within past 6 months) or glucocorticoids (within past 3 months), immediately stopped DMARDs and were monitored for 6 months. Participants were reviewed at 4, 12 and 24 weeks following DMARD cessation, plus additional patient-requested ad-hoc visits if flare occurred between these visits. Flare was defined as any single measure of DAS28-CRP > 2.4 during the follow-up period. Blood samples were collected at baseline (i.e. immediately prior to DMARD cessation) and at either flare onset or month 6 sustained DFR.

### Ethical approval

All participants provided written informed consent prior to participation. The BioRRA study protocol was approved by the North East–Tyne & Wear South Research Ethics Committee (National Health Service Health Research Authority, reference 14/NE/1042), and the study is registered at ClinicalTrials.gov (NCT02219347). The use of BioRRA study samples for the work presented in this manuscript was overseen by the Newcastle Biobank Committee under approval of the North East–Newcastle & North Tyneside 1 Research Ethics Committee (National Health Service Health Research Authority, reference 17/NE/0361).

### PBMC isolation

Blood samples were collected at baseline (i.e. immediately prior to DMARD cessation) and at either flare onset or month 6 sustained DFR. Healthy control PBMCs (technical biological replicates) were obtained at a single time point from a single healthy donor provided as a cone product by National Health Service Blood and Transplant. Blood was diluted in an equal volume of calcium/magnesium-free Hanks medium (Lonza, catalogue number BE10-543F) with 2 mM ethylenediaminetetraacetic acid (EDTA, Thermo Fisher Scientific, catalogue number BP2482100). Between 15–25 ml of diluted blood was then layered onto 15 ml of LymphoPrep™ (Axis-Shield Diagnostics, catalogue number NYC 1114547) and spun at 895 × $g$ for 30 min at room temperature with slow acceleration with no brake. PBMCs were recovered from the density interface by pipetting, and suspended in wash medium containing 50 ml of calcium/magnesium-free Hanks medium with 1% fetal bovine serum (FBS) (Thermo Fisher Scientific, catalogue number Gibco 10270-106; Biosera, catalogue number FB-1550/500; Labtech, catalogue number FCS-SA/500) at 4 °C. Samples were spun at 600 × $g$ for 7 min at 4 °C, the supernatant discarded, and the pellet resuspended in 50 ml wash medium at 4 °C. Samples were then spun at 250 × $g$ for 7 min at 4 °C, the supernatant discarded and the cells resuspended in 10 ml of wash medium and passed through a 70 µm nylon filter (Greiner bio-one, catalogue number 542070). The cell concentration was measured using a Burker counting chamber, and samples were then spun at 400 × $g$ for 7 min at 4 °C. The supernatant was discarded, and the cells resuspended in FBS with 10% Dimethyl sulfoxide (DMSO, Sigma-Aldrich, catalogue number D2650) in 1 ml aliquots of between 5–10 million cells and stored overnight at −80 °C before transfer to long-term storage at −150 °C.

### Mass cytometry

A 44-marker pan-PBMC mass cytometry panel was designed, incorporating CD45 barcoding to allow multiplexing of 5 samples within each batch (Supplementary Table 2). Antibodies conjugated to 89-yttrium and 209-bismuth were supplied by the manufacturer (Standard Biotools). In-house conjugation of CD45 antibody with palladium isotopes (Trace Sciences) was performed using the metal ion chelator isothiocyanobenzyl-EDTA (Dojindo, catalogue number 105394-74-9)[58]. Metal conjugation of other antibodies was performed using the Maxpar® x8 Multi-Metal Labelling Kit (Standard Biotools, catalogue number 201300) according to the manufacturer's instructions and including additional 113-indium, 115-indium and 157-gadolinium isotopes (Trace Sciences). PBMCs were thawed at 37 °C for 5 min, then suspended in 20 ml of thaw medium. The samples were then spun at 400 × $g$ for 8 min, the supernatant discarded and the pellet resuspended in 25 ml thaw medium at 37 °C. The samples were then spun again at 400 × $g$ for 8 min, the supernatant discarded, and the pellet resuspended in 2 ml of thaw medium and rested for 1 h at 37 °C. The cell concentration was measured using a Burker counting chamber. Following addition of 23 ml of thaw medium, the samples were spun at 400 × $g$ for 8 min, the supernatant discarded, and the pellet resuspended in 200 µl thaw medium per 1 million cells per well in a 96-well plate (Griener bio-one, catalogue number 651101). Subsequent wash steps were performed at 500 × $g$ for 5 min at room temperature using wash buffer containing calcium/magnesium-free Dulbecco phosphate-buffered saline (DPBS) (Sigma-Aldrich, catalogue number D8537) with 2% FBS. The cells were washed once, and stained with barcoding CD45 antibodies in wash buffer (total staining volume 50 µl per well) for 30 min at room temperature. The cells were then washed once, followed by incubation with 50 µl DPBS with 2.5 µM cisplatin (Standard Biotools, catalogue number 201064) for 5 min at room temperature. Cells were then washed twice, and pooled for extracellular antibody mastermix staining in wash buffer (total staining volume 100 µl per well) for 60 min at room temperature. Cells were washed twice with DPBS, followed by fixation in 100 µl eBioscience working fix buffer (Thermo Fisher Scientific, catalogue number 00-5523) with 100 µl of DPBS with 3.2% formaldehyde (TAAB Laboratories Equipment Ltd, catalogue number F017/3) for 30 min at room temperature. Cells were then spun at 500 × $g$ for 5 min, the supernatant discarded, and the cells resuspended in 200 µl eBioscience working perm buffer (Thermo Fisher Scientific, catalogue number 00-5523). This process was

repeated, followed by intracellular antibody mastermix staining in working perm buffer (total staining volume 100 μl per well) for 60 min at room temperature. The cells were then washed twice in DPBS, and then incubated in 200 μl of DPBS with 125 nM iridium (Standard Biotools, catalogue number 201192A) and 1.6% formaldehyde for 60 min at room temperature. Cells were then washed and stored in 200 μl wash buffer overnight at 4 °C before acquisition using a CyTOF mass cytometer (Helios, Standard Biotools) at 30 μl/min after calibrating against the manufacturer's tuning protocol.

Samples were acquired in 20 batches of 5 samples each—one technical replicate and 4 patient samples (i.e. paired samples from 2 patients per batch). Raw cytometer data were normalised using the *Normalizer* (v0.3) Matlab application of Finck et al.[59], and then manually gated using FlowJo (v10, BD Biosciences) to remove debris, dead cells, doublets, and calibration beads. A combined dump stain for CD15, CD66b and CD203c was used to remove granulocytes and basophils. Cleaned data were then deconvoluted to individual samples by manual gating of CD45 antibody staining. Representative images of mass cytometry data pre-processing are shown in Supplementary Fig. 16.

Further processing and analysis of deconvoluted mass cytometry data was performed in the R statistical environment (v3.4). Normalisation of inter-batch variation was achieved by reference to the technical replicate samples using the *Normalizebatch()* function of the *cydar*[60] package (v1.2.1). Further analysis was performed according to the published workflow of Nowicka et al.[61]. Unsupervised clustering of normalised data using *FlowSOM*[62] (version 1.10.0) and *ConsensusClusterPlus*[63] (v1.42.0) packages was performed, using a 15 × 15 self-organising map followed by reduction to 50 clusters. Manual merging of clusters was performed based upon similarity of canonical lineage marker expression.

### Single-cell RNA sequencing (scRNAseq)

PBMCs were thawed at 37 °C for 5 min and then suspended in 20 ml of thaw medium. The cells were washed twice in 25 ml thaw medium at 400 × g for 8 min, the cell concentration was measured using a Burker counting chamber, and then the cells were resuspended in 200 μl thaw medium per 1 million cells per well in a 96-well plate. Subsequent wash steps were performed at 400 × g for 3 min at room temperature using fluorescence-activated cell sorting (FACS) buffer containing calcium/magnesium-free DPBS with 0.5% bovine serum albumin (Sigma-Aldrich, catalogue number A2153), 1 mM EDTA (Sigma-Aldrich, catalogue number E7889) and 0.01% sodium azide (Sigma-Aldrich, catalogue number S2002). The cells were washed and then stained with fluorescent antibody mastermix (Supplementary Table 3) in FACS buffer with 200 ng polyclonal human IgG (Octagam, Octapharma Ltd) at 4 °C for 30 min (final staining volume 50 μl) and protected from light. The cells were washed twice, and then incubated with 20 μl of Via-Probe (BD Biosciences, catalogue number 555815) at 4 °C for 10 min and protected from light. The cells were then suspended in 1 ml FACS buffer and kept on ice for immediate fluorescence-activated cell sorting (FACS) using FACS Fusion and FACS Aria III cell sorters (BD Biosciences). After removal of debris, doublets and dead cells, three subsets were isolated from each sample: CD3⁺CD4⁺CD45RO⁺PD1ʰⁱ T cells, CD3⁺CD8⁺CD45RO⁺PD1ʰⁱ T cells, and CD19⁺ B cells. Representative images of FACS manual gating are shown in Supplementary Fig. 17.

Isolated cells were immediately processed according to the BD Rhapsody protocol, including staining with oligo-tagged surface antibodies (Supplementary Table 4), loading onto a Rhapsody cartridge (ratio 2:2:1 CD4⁺:CD8⁺:B cells), and cell lysis. Excellent viability (median(range) 85(81–87)%) was observed immediately prior to cartridge loading. Sample tags were used to combine 6 individual cell isolates from a single patient on each cartridge (two visit time points, three isolates per time point). Beads were then processed using the Rhapsody Cartridge Kit (BD Biosciences, catalogue number 633733), Rhapsody Cartridge Reagent Kit (BD Biosciences, catalogue number

633731) and the Ab-O Single-Cell Human Sample Multiplexing Kit (BD Biosciences, catalogue number 633781) according to the standard BD Rhapsody protocol with additional V(D)J steps and including supplementary primers (Integrated DNA Technologies UK Ltd, Supplementary Data 4) as per the manufacturer's instructions. Beads were stored at 4 °C for <4 months prior to library preparation, which was performed simultaneously across all samples using the Rhapsody cDNA Kit (BD Biosciences, catalogue number 633773), Rhapsody Targeted Amplification Kit (BD Biosciences, catalogue number 633774) and Rhapsody Immune Response Panel Hs Kit (BD Biosciences, catalogue number 633750) with additional custom primers (BD Biosciences, catalogue number 633770, Supplementary Data 5) based on transcriptional markers identified in our previous published work[5]. Five indexed libraries (antibody, mRNA, TCR, BCR, SMK sample tag) were prepared and sequenced in two pooled runs: NovaSeq 6000 S2 300 cycle (75 × 225 bp) + 10% PhiX (mRNA, TCR, BCR, SMK sample tag libraries), and NovaSeq 6000 S2 100 cycle (2 × 50 bp) + 20% PhiX (antibody library). This approach ensured the entirety of each individual library was within a single sequencing run, thus avoiding sequencing batch effects. BCR data were not available for 3 patients (1 flare, 2 remission) due to insufficient cDNA obtained during the library preparation process.

FASTQ files were uploaded to the SevenBridges Genomics platform and processed using the standard BD Rhapsody deconvolution, alignment and V(D)J sequencing pipelines according to the manufacturer's instructions (BD Rhapsody Targeted Analysis Pipeline, v1.0). The resulting feature count matrices were then processed in R (v4.1.0) according to the standard *Seurat* (v4.0)[64] workflow with default parameters unless otherwise specified. The weighted nearest neighbour method was to achieve unsupervised clustering of surface protein and mRNA features simultaneously. The *FindClusters* function was run with a resolution of 0.4 for CD4⁺ T cell data, and 0.3 for CD8⁺ T cell and B cell data. Clusters comprised of fewer than 100 cells were excluded from downstream analyses.

### Statistical analysis

For mass cytometry data, the statistical significance of differences in circulating proportional abundance of clusters between different sample types was achieved by use of a generalised linear mixed model, accounting for both overdispersion and sample pairing, with adjustment for multiple testing within each pairwise sample type contrast by Benjamini–Hochberg correction. Adjusted two-sided *p* values < 0.05 were deemed statistically significant.

For scRNAseq data, differential feature expression between clusters and paired patient group contrasts was assessed by the Wilcoxon rank sum test, with Bonferroni multiple test correction within each cluster and paired contrast, and a fold-change threshold of >±1.5. Comparison of proportional subset abundance was performed within each cell type (i.e. of total CD4⁺ T cells, CD8⁺ T cells, and B cells) by the Wilcoxon rank sum test, with Benjamini–Hochberg multiple test correction within each paired contrast. Clonal diversity was assessed by Shannon entropy, with significance testing using the Hutcheson *t*-test with Benjamini–Hochberg correction. Differential clone abundance within individual patients between paired visits was assessed by the exact Fisher test with Benjamini–Hochberg correction. A two-sided *p* < 0.05 after multiple test correction was deemed statistically significant.

### Reporting summary

Further information on research design is available in the Nature Portfolio Reporting Summary linked to this article.

## Data availability

The raw and processed scRNAseq data generated in this study have been deposited at the NCBI Gene Expression Omnibus under accession

code GSE245403. Mass cytometry cluster abundance (Supplementary Data 1), scRNAseq cluster abundance (Supplementary Data 2), scRNAseq cluster differential marker expression (Supplementary Data 3), scRNAseq library preparation additional primer sequences (Supplementary Data 4) and scRNAseq custom primer sequences (Supplementary Data 5) are provided as Supplementary Data and have been deposited at Zenodo[65] (https://doi.org/10.5281/zenodo.10507330). Source data are provided with this paper.

## Code availability
R analysis scripts are deposited at Zenodo[65] (https://doi.org/10.5281/zenodo.10507330).

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

## Acknowledgements

We would like to thank Thomas Otto (Glasgow University) for sharing R analysis scripts. This work was supported by grants from the Wellcome Trust (102595/Z/13/A to K.F.B.), National Institute for Health and Care Research (NIHR) Newcastle Biomedical Research Centre (BH136167/PD0045 to K.F.B.), British Society for Rheumatology (K.F.B.), Academy of Medical Sciences Starter Grant (supported by the Wellcome Trust, the Medical Research Council, the British Heart Foundation, Versus Arthritis, Diabetes UK and the British Thoracic Society Helen and Andrew Douglas bequest: SGL022\1074 to K.F.B.), Newcastle University Wellcome Trust Translational Partnership grant (K.F.B.), Newcastle Hospitals Charity grant (8033 to K.F.B.), Newcastle Health Innovation Partners Senior Clinical Fellowship (K.F.B.), and an NIHR Clinical Lectureship (CL-2017-01-004 to K.F.B.). Work in our laboratory is supported by the Research into Inflammatory Arthritis Centre Versus Arthritis (RACE) (grant number 20298), and Rheuma Tolerance for Cure (European Union Innovative Medicines Initiative 2, grant number 777357). The views expressed are those of the authors and not necessarily those of the NHS, the NIHR, or the Department of Health and Social Care. Preliminary data within this manuscript was previously published as an oral abstract at the 2022 EULAR Congress[66].

## Author contributions

Conceptualisation: KFB and JDI. Methodology: KFB, DMcD, GH, RH, JC, AGP, AF, AEA, DS, ARS, HEM, and LMacD. Investigation: KFB, DMcD, GH, RH, and AF. Formal analysis: KFB. Funding acquisition: KFB. Supervision: AGP and JDI. Writing—original draft: KFB, AEA, and JDI. Writing—review and editing: KFB, AEA, AGP, JDI, DMcD, GH, and AF.

## Competing interests

K.F.B., A.G.P. and J.D.I. are named as inventors on a patent application by Newcastle University ("Prediction of Drug-Free Remission in Rheumatoid Arthritis"; International Patent Application Number PCT/GB2019/050902). D.Mc.D., G.H., R.H., J.C., D.S., A.R.S., H.E.M., L.Mac.D., A.F. and A.E.A. have no competing interests to declare.
