## [Peer Review File · Nature Communications]

Single-cell insights into immune dysregulation in rheumatoid arthritis flare versus drug-free remissionREVIEWER COMMENTS

Reviewer #1 (Remarks to the Author):

In this paper, Baker et al employ high dimensionality approaches to define immunological signatures, which can discriminate patients with rheumatoid arthritis, in remission under csDMARDs, who relapse or not upon therapy withdrawal. The study piggybacks on the BioRRA study, thus inheriting its advantages and its limitations.

This study, as conducted and presented here, has significant merits and concerning drawbacks. I summarise the most relevant here, in the hopes to be useful:

Merits:

- The study aims at addressing a dramatic unmet medical need and could bridge a deep knowledge gap
- As such, the potential translational relevance is very high
- State of art, high dimensional technologies are concurrently employed, providing the necessary validation
- The sample studied is well characterised and homogeneous
- The research group is highly reputable

Drawbacks:

- Given the natural remitting-relapsing nature of RA, the BioRRA study design may not fully discriminate true remission from simple transient disease control, due to the relatively short observation window after withdrawal. While this limits the relevance of the findings, it would not be fair to ascribe the responsibility to the authors, but at least this limitation should be acknowledged and discussed
- The study design lacks fundamental controls, namely health subjects and disease controls. How can the findings be certainly disease and disease-status specific, as enthusiastically claimed, without such controls?
- The approach is not that novel. For instance, Leong et al (ARD 2019) used high dimensionality approaches to identify a signature predictive of relapse upon withdrawal from TNFi
- The whole manuscript is extremely descriptive, all mechanistic associations are by theoretical inference only
- There is no demonstration of any actual clinical relevance of any of the findings. A multiple

regression analysis should be performed

- The layout of some of the figures does not convey immediately the message as portrayed. DRG could be used, for instance, when depicting differential gene expression

In summary, the data as presented falls significantly short of its very important goals and does not support the enthusiastic claims. As such, this study can be an useful atlas of interest to a more restricted, specialised audience

Reviewer #2 (Remarks to the Author):

An important outstanding question in rheumatology is whether patients with RA in longstanding remission need to continue lifelong therapy or whether they can discontinue conventional synthetic disease modifying anti-rheumatic drugs (csDMARDs). Many patients who discontinue therapy eventually flare and predictors and mechanisms are unknown. Baker et al have collected an interesting cohort of 36 RA patients in remission who discontinued csDMARDs, of which, 20 flared within 6 months of follow up and 16 did not. In this study, they compared blood cell measurements of CytoF and scRNA at baseline compared to flare or the 6 month follow up visit in those who did not flare. The analysis focused on lymphocytes and revealed changes in abundance and or gene expression in baseline versus flare. Overall, this is an interesting, rich dataset with findings that could be strengthened by some additional organization and focus to streamline the figures and allow bigger, more readable fonts. Comments are below:

Table 1 Study cohort:

Several previous studies of treatment discontinuation have noted the duration of remission was predictive of likelihood of flare, with patients in prolonged remission less likely to flare. It would be helpful to know whether remission of a certain duration was an inclusion criterion for this study and how long the patients were in remission upon enrollment. This could be added to Table 1.

The authors note in the introduction that the strength of their study design was a controlled DMARD cessation providing a novel experimental medicine model to synchronise flare. It would be helpful to know the median time to flare, this also could be added to Table 1. I was surprised that they did not make an attempt to analyze their data according to time to flare,

but perhaps there was not enough power for this?

Figure 1 Mass Cytometry Data

Overall, this figure suffers from having text that is too small to read. One option is to move some of the data to supplement and focus on the box plots, which are useful to see the individual data points, and to me are a highlight of the paper. Specifically, I think it would be helpful to present all 8 cell types that were different over time in those with flare or sustained remission (CD4_₁, CD8_₂, CD8_₄, BC_₁, CD4_₃, GDT_₃, DC_₂, CD4_₂).

I recommend also continuing the nomenclature from Figure 1A and B in EFGH so the reader can connect the data with the heatmap and tSNE plot (ie BC_₁:CD19+CD27+CD86+CD21- B cells). This is a theme throughout the paper, as I will highlight below, since there are many different cell types and it is important to be as clear and consistent as possible to facilitate the reader to follow.

If you do want to keep the volcano plots, I recommend removing the vertical lines at $\text{Log}_2\text{FC}=\pm 1$ since you are not using this a threshold for significance, and it is distracting. (ie I originally was confused and thought only CD8_₄, CD8_₂, CD4_₃, BC_₁, CD4_₁ were significant but realize you are including GDT_₃ and DC_₂).

Of note, it is interesting that several of the cell types are increased in flare vs baseline and 6 mo remission vs baseline. Do the authors think this is a universal signature of DMARD withdrawal or could it mean the patients in 6 month remission that had increases in these cells are on their way to developing a flare.

Also, there is no mention of DC_₂, presumably since it decreases during flare, but to me that does not detract from it's potential significance.

Finally, it is important to note in the text that there were no differences detectable at baseline between those patients who go on to flare and those who do not as it underscores the complementarity of the scRNA data to come.

Figure 2 is a very nicely presented overview of the scRNA subsets. I would combine the terms on the right and left of the dot plots in D, E, F, to facilitate reading (ie CD4__A: undistinguished)

Figure 3 presents the important findings re the differences in proportional abundances in CD4 T cells between flare and baseline. Figure 3A has not stars calling out the significant differences in abundances. To better highlight significant findings, this figure could be filtered to present only the significant different cell types (CD4__F, CD4__I, CD4__J, CD4__P,

CD4_A and present them as a series of box plots (as in Figure 1EFGH). This would enable plotting them with their own y axis, so the reader can see the extent of the differences. It would be helpful to use the full name, ie CD4_F: CCR7/ICOS etc so the reader can more easily follow along.

The Radar plots in Figure 3BCD really belong conceptually with Figure 4 and to me it is confusing to mix the abundance data with the gene expression data this way.

Figure 4 presents a series of volcano plots with rich data but it suffers from being too small to read and a bit overwhelming. One suggestion is to organize the data to three groups (or lines) of comparisons:

- 1) DEG that are predictive of impending flare (FB v RB)
- 2) DEG that are characteristic of flare v baseline (FF v FB)
- 3) DEG that are characteristic of flare v sustained remission (FF v RS)

CD4_1, CD4_M, CD4_F, and CD4_B have DEG in FBvRB and FFvFB and could be placed on top of each other to enable these direct comparisons as well.

It would be better to call out fewer genes (perhaps just the ones highlighted in the text) and then allow them to have larger font.

Figure 5

Please clarify the writing in the description of the percentages of the clonality in the results. For example, it was not clear how you calculated 11.3% clonal CD4 T cells when 1,021 of 4,618 cells were from clonal pairs.

Figure 5A-C are not readable. It would be worth deciding what the key message is and making that text legible.

Figure D-F are effective visualizations. I wonder if these plots were colored as continuous variables with respect to the number of clones if this would further enhance the information on the plot. It looks like the CD4_B subset is really the most clonally expanded and I wonder if Figure D would then be a very dark shade of red, while Figure E might be a lighter shade of red?

One weakness is that there were 8 patients in the flare group and only 4 patients in the sustained remission group. Therefore there was more power to detect differences over time in the flare group compared to the sustained remission group.

The model in Figure 6 is helpful to synthesize the plethora of data. I would consider adding the term flare in the figure to clarify the message.

Reviewer #3 (Remarks to the Author):

This study by Baker, et al., identifies immune changes following rheumatoid arthritis (RA) flare triggered by removal of immunosuppressive therapies (compared to patients who do not experience a flare). Overall, this is a well-designed study and well-written manuscript that offers interesting insight into the immune changes identified at baseline and at the time of flare in RA patients. Fig. 6 provides a helpful, concise summary of the major findings.

Suggestions below would improve the manuscript by adding clarity about the data presented and incorporating additional analyses required to support some of the conclusions drawn. Added discussion of related previous studies (specified below) would also improve the contextualization of findings within the larger field of RA immunology.

Major points:

- Fig. 5A-C: Which patients/time points are shown in the circos plots? That is not clear from the legend. Similar comment for Fig. 5D-F; group(s) need to be clarified– flare patients only? Flare time point only? Minor related points: It would also be helpful to 1) add the labels across the top of each plot indicating which immune cell subset is plotted for Fig. 5A-C (i.e., the information that is currently only listed in the legend) and 2) add titles at the top of Fig. 5D-F to improve clarity.
- Fig. 5 states “...significant circulating T cell clonal expansion in flare but not in DFR patients,” but DFR patient data is not shown for reference, nor is it clear how that claim was statistically supported. Clonally expanded cells should be shown on UMAPs for each of your four groups (FB, FF, RB, RS) and comparative data needs to be added to that figure with appropriate statistical comparisons (between flare and remission patients, as the legend title suggests). Perhaps the number of individual T cell clonotypes that are expanded can be plotted per patient and then statistically compared across groups. Same for contracted T

cells. Or perhaps you could quantify the extent of clonal expansion per patient in some other way.

- Immune repertoire findings (T cell clonal expansion in flare, Fig. 5) are quite interesting but are not mentioned in the abstract.
- IgA class switching involves deletion of the IgD constant region-coding DNA. Do the authors have an explanation for the rise in IgD transcript (line 216)? Perhaps this reflects a rise in antibody-secreting cells that have recently undergone class-switch, explaining the temporary presence of both transcripts?
- Expansion of CD21^{lo}/neg B cells has previously been noted in RA (PMC3373152); this paper should be cited.
- A paper by Zhang et al uses single-cell and CyTOF to examine inflammatory immune signatures in RA synovium (PMC6602051); this paper should be cited/discussed.
- “TBC” is listed for a data repository location and for the link to analysis code. In the manuscript, authors state the analysis code is available upon reasonable request from the senior author (line 495), which seems to contradict the Nat. Comm. form stating that code will also be deposited). Please adjust either manuscript text or the Nat. Comm form to make them consistent with one another. Related to this, the data need to be deposited with a valid link prior to publication, if not the analysis code as well. I presume this will be corrected prior to publication (note: I received a zip file containing these materials).

Minor points:

- You may want to make the point in the introduction that peripheral blood markers are more translatable to clinical testing, unlike tissue biomarkers which are difficult to repeatedly assay.
- The Fig. 4 legend title makes it sound like you are combining transcriptomic and

phenotypic changes in the same volcano plot, but what I think you mean is that you used phenotypic markers to define the subsets (top title of each plot), but are only showing DEGs in the volcano plots, correct? I would suggest you just remove the phrase “surface protein expression” from the figure legend title for clarity.

- The current abbreviations are correctly defined (FB, FF, RB, RS), but when I see “R”, my brain keeps toggling between remission (what you defined) and relapse (the opposite of what is meant). Perhaps other readers will not have this issue, but what about FlareB and RemB to avoid this? I’m also struggling a tad with FF and RS both referring to the subsequent (non-baseline visit) yet having a different second letter. What about FV, flare visit and RV, remission visit? These are just suggestions.

- TCF7 (TCF-1) has a wide array of functions in CD4+ cells (e.g., PMID: 34127847), including those related to Tfh differentiation. Given the reduction in CXCR5 with flare onset, has the role of Treg conversion to Tfh been considered, as this process has been noted in disease, albeit in an animal model and different inflammatory process (PMC5854619)?

- To support your IgA plasma cell link in the discussion, it may be helpful to cite/discuss some of the following papers:

- o IgA recognition of gut commensals in human RA (PMC9804515)

- o Ggut commensals are required to incite RA-like disease in the K/BxN mouse model (PMC2904693)

- o Changes in IgA recognition of commensals impact arthritis severity (PMC9002138)

Response to reviewer comments

We would like to thank the reviewers for their very helpful feedback. We have incorporated their suggestions in the revised version of our manuscript, and provide an itemised response to their comments below.

During the process of revising the manuscript, we discovered an issue with our analysis which had resulted in an unintentional switch of outcome (i.e. flare vs DFR) for some patients in the single cell sequencing experiment. Whilst the mass cytometry data and scRNAseq clustering remain unchanged, there was modest impact upon the differential abundance and gene expression analyses of the scRNAseq data, and minimal impact upon the BCR/TCR analyses. However, the key findings and conclusions of our work remain the same. We sincerely apologise for this inadvertent oversight - we have now corrected the problem in our analysis, and updated the manuscript and figures as appropriate.

Where page and line numbers are referenced in our responses, we refer to the tracked changes version of the revised manuscript.

Reviewer 1: Comment 1

In this paper, Baker et al employ high dimensionality approaches to define immunological signatures, which can discriminate patients with rheumatoid arthritis, in remission under csDMARDs, who relapse or not upon therapy withdrawal. The study piggybacks on the BioRRA study, thus inheriting its advantages and its limitations.

This study, as conducted and presented here, has significant merits and concerning drawbacks. I summarise the most relevant here, in the hopes to be useful:

Merits:

- The study aims at addressing a dramatic unmet medical need and could bridge a deep knowledge gap*
- As such, the potential translational relevance is very high*
- State of art, high dimensional technologies are concurrently employed, providing the necessary validation*
- The sample studied is well characterised and homogeneous*
- The research group is highly reputable*

Drawbacks:

- Given the natural remitting-relapsing nature of RA, the BioRRA study design may not fully discriminate true remission from simple transient disease control, due to the relatively short observation window after withdrawal. While this limits the relevance of the findings, it would not be fair to ascribe the responsibility to the authors, but at least this limitation should be acknowledged and discussed*

Authors' response

We agree. Patients were followed closely in the BioRRA study at 1, 3 and 6 months following DMARD cessation (with additional ad-hoc visits at patient request), and thus we are confident that clinical remission was maintained during study follow-up. However, our observations are limited by the 6-month follow up duration of the study, and we do not have outcome data beyond this time. We did touch upon this limitation in the discussion of our original manuscript, but have now highlighted this further and made more explicit in the revised version (page 24, lines 519-523).

Reviewer 1: Comment 2

The study design lacks fundamental controls, namely health subjects and disease controls. How can the findings be certainly disease and disease-status specific, as enthusiastically claimed, without such controls?

Authors' response

We thank the reviewer for their comment. Whilst the reviewer highlights an important issue, we believe that inclusion of such controls would only be relevant for work downstream of this report.

Firstly, we would like to emphasise that the design of the BioRRA study is such that control subjects are not required for direct comparisons with remission and flare patients. As a longitudinal cohort study, all patients are subjected to the same intervention (i.e. DMARD cessation), and during the follow-up period subsequently differentiate into two groups – remission and flare. In this sense, the remission group acts as the requisite (and sufficient) controls for the flare group. Inclusion of additional control groups, either healthy controls or early/established RA patients with active disease, is not required to draw comparisons between patients who flare versus those who remain in remission following DMARD cessation, which is the central aim of this manuscript.

Our observations provide evidence for specific immune cell subsets which differ in abundance and/or transcriptional profile between flare and remission groups, and as such may be suitable for further translational biomarker studies. Such biomarkers could be applied to the prediction of flare prior to DMARD cessation, or the monitoring for flare following DMARD cessation. During future development of such biomarkers, it would be important to compare their levels across healthy volunteers and also patients on continued DMARD therapy, both for biomarker validation and calibration purposes. It would also be of interest to compare with other immune-mediated inflammatory diseases to ascertain those which may be disease-specific versus those which may be more generalizable to an inflammatory state. However, we respectfully argue that such analyses – whilst important for biomarker development and other downstream work– are

not relevant to the direct comparison of flare versus remission following DMARD cessation within the BioRRA study, this being the central purpose of our manuscript.

In acknowledgement of the reviewer's important comment, we have added further text to the discussion section to emphasise the lack of healthy and disease controls, and to highlight that we are thus not able to generalise our findings outside the context of the BioRRA study cohort (page 24, lines 524-529). We hope the discussion above and addition to the manuscript text satisfactorily addresses the reviewer's concerns.

Reviewer 1: Comment 3

The approach is not that novel. For instance, Leong et al (ARD 2019) used high dimensionality approaches to identify a signature predictive of relapse upon withdrawal from TNFi

Authors' response

Agreed – whilst we are not aware of another published study of single-cell high-dimensionality analyses using a model of complete DMARD cessation to study drug-free remission in RA, we accept that the practice of DMARD cessation more generally is not novel. We have now removed the word "novel" from descriptions of our experimental approach, and have also included reference to the juvenile idiopathic arthritis study of Leong et al in our discussion (page 23, lines 508-510).

Reviewer 1: Comment 4

The whole manuscript is extremely descriptive, all mechanistic associations are by theoretical inference only

Authors' response

We aimed to present exploratory observational data within this manuscript, and agree that the term "descriptive" is therefore quite appropriate. Whilst we consider our findings generate hypotheses that warrant further study, we did not intend to claim mechanistic proof beyond the observations reported. We have now clarified this further in our discussion (page 24, lines 529-531).

Reviewer 1: Comment 5

There is no demonstration of any actual clinical relevance of any of the findings. A multiple regression analysis should be performed

Author's response

As mentioned above, in writing this manuscript we aimed to provide a descriptive overview of the immunological features that differentiate flare from DFR in RA. Whilst

these observations and cell subsets identified offer potential avenues for future clinical biomarker development, we are unable to explore this further within our limited scRNAseq cohort (12 patients) as we do not have sufficient statistical power for multiple regression analysis. We have added this as a limitation of the study in our discussion (page 23, lines 510-511).

Reviewer 1: Comment 6

The layout of some of the figures does not convey immediately the message as portrayed. DRG could be used, for instance, when depicting differential gene expression

Authors' response

Please accept our apologies, but we are not sure of the meaning of "DRG" in this context. We hope the individual figure amendments suggested by reviewers 2 and 3 as answered below have addressed these concerns, but we would be happy to address any further specific issues with the figures with further clarification.

Reviewer 1: Comment 7

In summary, the data as presented falls significantly short of its very important goals and does not support the enthusiastic claims. As such, this study can be an useful atlas of interest to a more restricted, specialised audience

Authors' response

We thank the reviewer for this comment, which encapsulates many of their previous comments above. Our intention in writing this manuscript was to provide a descriptive overview and starting point for further translational research. However, reflecting on the reviewer's comments we do concede that the wording of our manuscript, especially in the title, abstract, aims and final conclusions, does perhaps infer an over-enthusiastic claim in regards to the clinical relevance of our findings. We agree with the reviewer's suggestion that the primary purpose of this manuscript is to provide an atlas and single cell dataset for the study of flare and drug-free remission in rheumatoid arthritis. In response, we have emphasised this point throughout the revised manuscript by making the following changes:

1. We have changed the title of the manuscript to shift emphasis towards descriptive/observational data rather than mechanistic inferences
2. We have reworded the final sentence of the abstract to de-emphasise the implications of our results for current clinical practice

3. We have reworded the final paragraph of the introduction, again to de-emphasise the implications of our results for current clinical practice (page 4, lines 72-74)
4. We have reworded the final paragraph of the discussion section to make explicit that we provide observational data only, which requires further translational research before application to future clinical practice (pages 24-25, lines 532-543)

Reviewer #2 (Remarks to the Author):

Reviewer 2: Comment 1

An important outstanding question in rheumatology is whether patients with RA in longstanding remission need to continue lifelong therapy or whether they can discontinue conventional synthetic disease modifying anti-rheumatic drugs (csDMARDs). Many patients who discontinue therapy eventually flare and predictors and mechanisms are unknown. Baker et al have collected an interesting cohort of 36 RA patients in remission who discontinued csDMARDs, of which, 20 flared within 6 months of follow up and 16 did not. In this study, they compared blood cell measurements of CytoF and scRNA at baseline compared to flare or the 6 month follow up visit in those who did not flare. The analysis focused on lymphocytes and revealed changes in abundance and or gene expression in baseline versus flare. Overall, this is an interesting, rich dataset with findings that could be strengthened by some additional organization and focus to streamline the figures and allow bigger, more readable fonts. Comments are below:

Table 1 Study cohort:

Several previous studies of treatment discontinuation have noted the duration of remission was predictive of likelihood of flare, with patients in prolonged remission less likely to flare. It would be helpful to know whether remission of a certain duration was an inclusion criterion for this study and how long the patients were in remission upon enrollment. This could be added to Table 1.

Authors' response

The design of the BioRRA study did not include a run-in observation period prior to DMARD cessation, and thus DAS28-CRP data are not available to calculate the duration of remission prior to enrolment. However, "time since last change in DMARD therapy" and "time since last systemic/intra-articular steroid" were recorded at enrolment as surrogate measures of disease stability, and are included in Table 1. Additionally, the BioRRA study inclusion criteria required that patients must not have received enteral, parenteral or intra-articular steroids within 3 months of enrolment – this has now been added to the methods section (page 25, lines 552-553).

Reviewer 2: Comment 2

The authors note in the introduction that the strength of their study design was a controlled DMARD cessation providing a novel experimental medicine model to synchronise flare. It would be helpful to know the median time to flare, this also could be added to Table 1. I was surprised that they did not make an attempt to analyze their data according to time to flare, but perhaps there was not enough power for this?

Authors' response

Demographic data for the BioRRA cohort is detailed in the original study publication (PMID: 31280933). For ease of reference and further clarity, we also show demographic details for the participants that comprise the mass cytometry and scRNAseq components of this manuscript in Table 1, and have now added time-to-flare to Table 1 as suggested. The relatively small sample size of this cohort, especially for the single cell sequencing analysis, prevents a meaningful analysis of time to flare using techniques such as Cox regression due to a lack of statistical power.

Reviewer 2: Comment 3

Figure 1 Mass Cytometry Data

Overall, this figure suffers from having text that is too small to read. One option is to move some of the data to supplement and focus on the box plots, which are useful to see the individual data points, and to me are a highlight of the paper. Specifically, I think it would be helpful to present all 8 cell types that were different over time in those with flare or sustained remission (CD4_, CD8_2, CD8_4, BC_1, CD4_3, GDT_3, DC_2, CD4_2).

Authors' response

We have revised Figure 1 by:

1. Moving the volcano plots to Supplementary Figure S1
2. Adding boxplots to include all 8 cell types as suggested above
3. Ordering the boxplots (Figs 1B-I) to match the order of the cluster key in Figure 1A to aid readability
4. Increasing the font size for the cluster names in Fig 1A

Reviewer 2: Comment 4

I recommend also continuing the nomenclature from Figure 1A and B in EFGH so the reader can connect the data with the heatmap and tSNE plot (ie BC_1:CD19+CD27+CD86+CD21- B cells). This is a theme throughout the paper, as I will highlight below, since there are many different cell types and it is important to be as clear and consistent as possible to facilitate the reader to follow.

Authors' response

The cluster name as per Figure 1A is now included in the labels for the boxplots as suggested.

Reviewer 2: Comment 5

If you do want to keep the volcano plots, I recommend removing the vertical lines at $\text{Log}_2\text{FC} = \pm 1$ since you are not using this a threshold for significance, and it is distracting. (ie I originally was confused and thought only CD8_4, CD8_2, CD4_3, BC_1, CD4_1 were significant but realize you are including GDT_3 and DC_2).

Authors' response

Vertical lines on volcano plots have now been removed (figures now moved to supplementary Figure S2).

Reviewer 2: Comment 6

Of note, it is interesting that several of the cell types are increased in flare vs baseline and 6 mo remission vs baseline. Do the authors think this is a universal signature of DMARD withdrawal or could it mean the patients in 6 month remission that had increases in these cells are on their way to developing a flare.

Authors' response

We agree that such longitudinal changes in abundance that are common to both flare and remission patients could represent an effect of DMARD withdrawal. We have made this more explicit with an additional sentence in the results section (pages 4-5, lines 88-89). It is also possible that some of the patients in remission at month 6 may have subsequently experienced an arthritis flare, but we unfortunately do not have longer-term follow-up data to investigate this further.

Reviewer 2: Comment 7

Also, there is no mention of DC_2, presumably since it decreases during flare, but to me that does not detract from its potential significance.

Authors' response

Agreed – we have now included the reduction in DC_2 proportional abundance at flare onset within the text of the results section (page 5, lines 95-97), and also now include the boxplot of this data as part of the abovementioned revisions to Figure 1.

Reviewer 2: Comment 8

Finally, it is important to note in the text that there were no differences detectable at baseline between those patients who go on to flare and those who do not as it underscores the complementarity of the scRNA data to come.

Authors' response

Agreed – we have added this to the end of the mass cytometry results section (page 5, lines 102-104)

Reviewer 2: Comment 9

Figure 2 is a very nicely presented overview of the scRNA subsets. I would combine the terms on the right and left of the dot plots in D, E, F, to facilitate reading (ie CD4_A: undistinguished)

Authors' response

The cluster names and phenotype have now been combined together on the right side of the dot plots as suggested.

Reviewer 2: Comment 10

Figure 3 presents the important findings re the differences in proportional abundances in CD4 T cells between flare and baseline. Figure 3A has not stars calling out the significant differences in abundances. To better highlight significant findings, this figure could be filtered to present only the significant different cell types (CD4_F, CD4_I, CD4_J, CD4_P, CD4_A and present them as a series of box plots (as in Figure 1EFGH). This would enable plotting them with their own y axis, so the reader can see the extent of the differences. It would be helpful to use the full name, ie CD4_F: CCR7/ICOS etc so the reader can more easily follow along.

Authors' response

As suggested, Figure 3 has now been revised to show boxplots only for clusters with significant differences in abundances according to our updated analysis, using the full cluster name. The summary boxplot of all CD4+ T cells clusters has been retained within supplementary figure S3 (with addition of asterisks to indicate statistical significance).

Reviewer 2: Comment 11

The Radar plots in Figure 3BCD really belong conceptually with Figure 4 and to me it is confusing to mix the abundance data with the gene expression data this way.

Authors' response

The radar plots have now been moved to Figure 4 as suggested.

Reviewer 2: Comment 12

Figure 4 presents a series of volcano plots with rich data but it suffers from being too small to read and a bit overwhelming. One suggestion is to organize the data to three groups (or lines) of comparisons:

- 1) DEG that are predictive of impending flare (FB v RB)*
- 2) DEG that are characteristic of flare v baseline (FF v FB)*
- 3) DEG that are characteristic of flare v sustained remission (FF v RS)*

CD4_1, CD4_M, CD4_F, and CD4_B have DEG in FBvRB and FFvFB and could be placed on top of each other to enable these direct comparisons as well.

Authors' response

We have restructured the layout of Figure 4, taking into account your comments above and our updated analysis. To simplify the figure we have reduced the number of panels and now present two rows of comparisons: FlareB vs RemB on the first row, and FlareV vs RemV on the second row. Labels have been added to each row to help guide the reader. Additional volcano plots have been moved to Supplementary Figure S4 where needed.

Reviewer 2: Comment 13

It would be better to call out fewer genes (perhaps just the ones highlighted in the text) and then allow them to have larger font.

Authors' response

The volcano plots in Figure 4 have been adjusted to call out only those markers highlighted in the text, and to increase the font size of the marker names.

Reviewer 2: Comment 14

Figure 5

Please clarify the writing in the description of the percentages of the clonality in the results. For example, it was not clear how you calculated 11.3% clonal CD4 T cells when 1,021 of 4,618 cells were from clonal pairs.

Authors' response

Apologies – we were attempting to describe the number of unique clonal CDR3 sequences, the number of cells which expressed them, and the percentage of total cells that were clonal cells. The text has now been revised to clarify this (page 15, lines 333-336).

Reviewer 2: Comment 15

Figure 5A-C are not readable. It would be worth deciding what the key message is and making that text legible.

Authors' response

The circular plots are intended to provide the reader with a high-level graphical representation of the clonal diversity within each cell type (analogous to a tsNE/UMAP plot of unsupervised cell clustering). As they are not essential to the manuscript we have now moved these plots to the supplementary figures S5-7, with larger size for each image to improve readability.

Reviewer 2: Comment 16

Figure D-F are effective visualizations. I wonder if these plots were colored as continuous variables with respect to the number of clones if this would further enhance the information on the plot. It looks like the CD4_B subset is really the most clonally expanded and I wonder if Figure D would then be a very dark shade of red, while Figure E might be a lighter shade of red?

Authors' response

Agreed – we have set a transparency on the shading of the plots ($\alpha = 0.3$ for highlighted cells, and 0.2 for background cells) to create this effect, such that the greater the overlap (density) of clonal cells, the more intense the red shading (now Fig. 5A-C).

Reviewer 2: Comment 17

One weakness is that there were 8 patients in the flare group and only 4 patients in the sustained remission group. Therefore there was more power to detect differences over time in the flare group compared to the sustained remission group.

Authors' response

Agreed – this is a weakness of the study, and applies to all analyses presented. This has been added to the limitations section in the discussion (page 23, lines 511-514).

Reviewer 2: Comment 18

The model in Figure 6 is helpful to synthesize the plethora of data. I would consider adding the term flare in the figure to clarify the message.

Authors' response

We have added labels underneath the see-saw figures to indicate factors that favour flare and remission respectively. We have also labelled the two parts as A and B to aid readability and to explicitly link to the explanation of the model in the figure legend.

Reviewer #3 (Remarks to the Author):

Reviewer 3: Comment 1

This study by Baker, et al., identifies immune changes following rheumatoid arthritis (RA) flare triggered by removal of immunosuppressive therapies (compared to patients who do not experience a flare). Overall, this is a well-designed study and well-written manuscript that offers interesting insight into the immune changes identified at baseline and at the time of flare in RA patients. Fig. 6 provides a helpful, concise summary of the major findings.

Suggestions below would improve the manuscript by adding clarity about the data presented and incorporating additional analyses required to support some of the conclusions drawn. Added discussion of related previous studies (specified below) would also improve the contextualization of findings within the larger field of RA immunology.

Major points:

Fig. 5A-C: Which patients/time points are shown in the circos plots? That is not clear from the legend. Similar comment for Fig. 5D-F; group(s) need to be clarified– flare patients only? Flare time point only? Minor related points: It would also be helpful to 1) add the labels across the top of each plot indicating which immune cell subset is plotted for Fig. 5A-C (i.e., the information that is currently only listed in the legend) and 2) add titles at the top of Fig. 5D-F to improve clarity.

Authors' response

The circular plots and UMAPs are intended only to provide a general high-level graphical overview of the clonal diversity and clone cluster distribution of our dataset respectively. As such, they include cells from all patients at all time points, and following your comment we have now made this explicit in the figure legend. We have also added titles to the figures as suggested. Following comments from reviewer 2 above, the circular plots have now been moved to supplementary figure S5-7, also allowing a larger image size for easier viewing.

Please note that these figures are not designed for displaying differences in clonality between different patients, time-points or patient groups. However, following your

comments we have now included UMAPs of clonal cell distribution per patient group in Supplementary Figures S8-10 (see also response to "Reviewer 3: comment 2" below). We did also create individual circular plots for different patient groups, though this generated multiple figures with rather subtle differences between them which were not particularly informative and hence have not been included in the manuscript (example shown below for circular plots for CD4+ T cell data for FlareB, FlareV, RemB and RemV groups).

CD4: Flare B

CD4: FlareV

CD4: RemB

CD4: RemV

Reviewer 3: Comment 2

Fig. 5 states "...significant circulating T cell clonal expansion in flare but not in DFR patients," but DFR patient data is not shown for reference, nor is it clear how that claim was statistically supported. Clonally expanded cells should be shown on UMAPs for each of your four groups (FB, FF, RB, RS) and comparative data needs to be added to that figure with appropriate statistical comparisons (between flare and remission patients, as the legend title suggests). Perhaps the number of individual T cell clonotypes that are expanded can be plotted per patient and then statistically compared across groups. Same for contracted T cells. Or perhaps you could quantify the extent of clonal expansion per patient in some other way.

Authors' response

We have now added UMAPs showing the distribution of clonal cells within different patient groups in Supplementary Figure S8-10. However, as clonal expansion occurs at an intra-patient (rather than inter-patient) level, we felt it is more appropriate to test the significance of clonal expansion at an individual patient level. Adopting a similar approach to previously published studies (e.g. Penkava *et al* (2020). Nat Commun; 11:4767) and as described in the methods section, we compared the differential proportional abundance of individual clones within individual patients between paired time points (i.e. FlareB vs FlareV for flare patients, and RemB vs RemV for DFR patients) using Fisher's exact test with Benjamini-Hochberg multiple test correction. As discussed in the results section and following our updated analysis, significant changes in clonal abundance were observed in 4 patients (2 flare, 2 DFR) for CD4+ T cells and 4 patients (2 flare, 2 DFR) for CD8+ T cells – figures showing this are included in the supplementary material (Figures S12-S13). As described in the results section, no significant changes in clonal abundance were observed for B cells. Clones which showed a significant (adjusted $p < 0.05$) difference in proportional abundance between paired visits were then taken forward for further exploration as to cluster distribution, as per Figure 5D-E (individual patient level) and Supplementary Figures S14-S15 (individual patient and clone level).

To further describe the number of significantly expanded/contracted clones at an individual patient level, we have added a new Supplementary Table S1 which we hope helps to clarify this further. We have also revised the legend for Figure 5 to clarify that panels D and E depict only those clonal cells which had significant longitudinal changes in proportional abundance when comparing before and after DMARD cessation.

Reviewer 3: Comment 3

Immune repertoire findings (T cell clonal expansion in flare, Fig. 5) are quite interesting but are not mentioned in the abstract.

Authors' response

We have now included a summary of the TCR/BCR results in the abstract (page 2, lines 36-39).

Reviewer 3: Comment 4

IgA class switching involves deletion of the IgD constant region-coding DNA. Do the authors have an explanation for the rise in IgD transcript (line 216)? Perhaps this reflects a rise in antibody-secreting cells that have recently undergone class-switch, explaining the temporary presence of both transcripts?

Authors' response

In the updated analysis, plasma cell IgD expression is reduced at flare relative to DFR (i.e. FlareV vs RemV). We agree that your comment above remains a plausible explanation for this observation, which has now been added to the results text (page 14, line 297-298).

Reviewer 3: Comment 5

Expansion of CD21lo/neg B cells has previously been noted in RA (PMC3373152); this paper should be cited.

Authors' response

This reference has now been added (page 19, line 421).

Reviewer 3: Comment 6

A paper by Zhang et al uses single-cell and CyTOF to examine inflammatory immune signatures in RA synovium (PMC6602051); this paper should be cited/discussed.

Authors' response

This reference has now been added and discussed (page 17-18, lines 379-384).

Reviewer 3: Comment 7

"TBC" is listed for a data repository location and for the link to analysis code. In the manuscript, authors state the analysis code is available upon reasonable request from the senior author (line 495), which seems to contradict the Nat. Comm. form stating that code will also be deposited). Please adjust either manuscript text or the Nat. Comm form to

make them consistent with one another. Related to this, the data need to be deposited with a valid link prior to publication, if not the analysis code as well. I presume this will be corrected prior to publication (note: I received a zip file containing these materials).

Authors' response

Raw RNA sequencing data has now been deposited with NCBI Gene Expression Omnibus (GEO), with accession number GSE245403. The record is currently private, and will be made publicly available upon publication of the article. A secure access token has been supplied to the Editorial Office if you wish to view the submission record. If the manuscript is accepted for publication, the final version of the analysis code will be uploaded to Zenodo and the DOI will be provided in the published manuscript. The manuscript text has now been updated to reflect this.

Reviewer 3: Comment 8

Minor points:

You may want to make the point in the introduction that peripheral blood markers are more translatable to clinical testing, unlike tissue biomarkers which are difficult to repeatedly assay.

Authors' response

We have now added this to the introduction (page 3, lines 56-58).

Reviewer 3: Comment 9

The Fig. 4 legend title makes it sound like you are combining transcriptomic and phenotypic changes in the same volcano plot, but what I think you mean is that you used phenotypic markers to define the subsets (top title of each plot), but are only showing DEGs in the volcano plots, correct? I would suggest you just remove the phrase "surface protein expression" from the figure legend title for clarity.

Authors' response

Figure 4 does combine both gene expression (mRNA) and surface protein expression (oligo-tagged AbSeq antibodies) as quantified in the single cell sequencing experiment. For clarity we have changed the colouring of the volcano plots so that transcriptomic markers are shown in red, and surface protein markers in blue.

Reviewer 3: Comment 10

The current abbreviations are correctly defined (FB, FF, RB, RS), but when I see "R", my brain keeps toggling between remission (what you defined) and relapse (the opposite of what is meant). Perhaps other readers will not have this issue, but what about FlareB and RemB to avoid this? I'm also struggling a tad with FF and RS both referring to the subsequent (non-baseline visit) yet having a different second letter. What about FV, flare visit and RV, remission visit? These are just suggestions.

Authors' response

We have changed FB, FF, RB and RS to FlareB, FlareV, RemB and RemV respectively on all figures as suggested.

Reviewer 3: Comment 11

TCF7 (TCF-1) has a wide array of functions in CD4+ cells (e.g., PMID: 34127847), including those related to Tfh differentiation. Given the reduction in CXCR5 with flare onset, has the role of Treg conversion to Tfh been considered, as this process has been noted in disease, albeit in an animal model and different inflammatory process (PMC5854619)?

Authors' response

Thank you for the suggestion. However, in the updated analysis there is no significant difference in CXCR5 expression in the CD4_I subset at flare (i.e. FlareV vs RemV, or FlareV vs FlareB), and hence this text has been removed from the revised manuscript.

Reviewer 3: Comment 12

To support your IgA plasma cell link in the discussion, it may be helpful to cite/discuss some of the following papers:

- o IgA recognition of gut commensals in human RA (PMC9804515)*
- o Gut commensals are required to incite RA-like disease in the K/BxN mouse model (PMC2904693)*
- o Changes in IgA recognition of commensals impact arthritis severity (PMC9002138)*

Authors' response

Thank you for the suggested references, these have now been included in the manuscript (page 20, lines 442-445).

Further authors' comments

In addition to the response to the reviewers' comments above, we have also made some additional revisions to the manuscript as follows:

1. Inclusion of additional relevant references which have newly emerged since original manuscript submission:
 - a. Povoleri et al (2023). Psoriatic and rheumatoid arthritis joints differ in the composition of CD8+ tissue-resident memory T cell subsets. *Cell Rep*; 42:112514.
 - b. Reijm S et al (2023). Autoreactive B cells in rheumatoid arthritis consist of activated CXCR3+ memory B cells and plasmablasts. *bioRxiv*, 2023.2005.2019.538699.
2. Minor adjustments to supplementary figure font sizes to aid readability
3. Inclusion of a supplementary data file, providing lists of differentially expressed markers within each scRNAseq cell subset for each pairwise group contrast. This provides the opportunity to explore expression of specific markers of interest within our analysis, making it more accessible to the reader (data which would otherwise require the reader to re-run our analysis code afresh).

REVIEWERS' COMMENTS

Reviewer #1 (Remarks to the Author):

This reviewer thanks the authors for their effort in addressing the various points raised. However, such effort is limited to merely adding some points in the text, without providing any additional data or arguments to address this reviewer's previous concerns. It is actually quite telling that, in their point by point answers, the authors agree with most of such concerns.

As a consequence, my opinion has not changed. This paper in its current configuration is descriptive, does not provide the necessary information to be impactful, lacks indispensable comparators and its novelty is limited. As suggested before, this can be a useful atlas-type resource paper for a more specialised journal

Reviewer #2 (Remarks to the Author):

An important outstanding question in rheumatology is whether patients with RA in longstanding remission need to continue lifelong therapy or whether they can discontinue conventional synthetic disease modifying anti-rheumatic drugs (csDMARDs). Many patients who discontinue therapy eventually flare and predictors and mechanisms are unknown. Baker et al have collected an interesting cohort of 36 RA patients in remission who discontinued csDMARDs, of which, 20 flared within 6 months of follow up and 16 did not. In this study, they compared blood cell measurements of CyToF and scRNA (on a subset 8 patients who flared and 4 who did not) at baseline compared to flare or the 6 month follow up visit in those who did not flare. The analysis focused on lymphocytes and revealed changes in abundance and or gene expression in baseline versus flare. The previous submission has been effectively reorganized and focused to streamline the figures and allow bigger, more readable fonts.

The noteworthy results are:

1. Mass cytometry of samples during flare harbor increased i) CD19+CD27+CD86+CD21- B cells, ii) CD4+CD45RO+ICOS+PD1+CD38hi T cells, and iii) increased CD8+CD45RO+, PD1hiCD38hi T cells. But no differences at baseline between those who went on to flare and

those who did not.

2. CITE-Seq of samples during flare harbor increased Foxp3+CD25+CTLA4+Helios+ CD4+T cells, ii) Granzyme+CD38+ CD4+T cells, iii) Proliferating CD4+T cells, iii) HLADR+CD38+CD8 T cells, iv) CXCR5+ CD8 T cells, v) IgD+CD24+ B cells, and vi) CXCR3+ B cells.

3. The most significant result is the data presented in Figure 4, which identified proteins and RNA within cell types that predicted future flare or sustained remission.

4. They also found DMARD cessation is followed by clonal expansion of CD4 and CD8 T cells.

5. They propose a model whereby Tregs of patients who are going to flare are dysfunctional and during flare, IgA plasma cells, cytotoxic CD4 and CXCR5+CD8 expand.

6. While cross sectional studies of RA flare are not difficult to collect, since patients come to clinical attention, samples from patients with impending flare are much more challenging to come by. The strength of this study is the finding in Figure 4, identified proteins and RNA within cell types that predicted future flare or sustained remission. The weakness of this data is that it is a rather small sample size and as reviewer 1 mentioned, we don't know if the patients sustained remission for longer than 6 months. The authors have addressed this in their discussion.

Reviewer #3 (Remarks to the Author):

This is a revised manuscript from Baker, et al., which highlights immune changes that followed removal of immunosuppressive therapies in patients who flared (compared to patients who did not flare). In my opinion, the data are of high interest to the RA community and their somewhat descriptive nature do not detract from their significance. This rich dataset should be of value to the larger rheumatology community.

Authors were generally responsive to reviewer comments and data are now presented with improved clarity. Caveats are now appropriately acknowledged, suggested citations were added/discussed, and the Discussion is improved.

Minor comments (revision suggested, but not required):

- Fig. S8, S9, S10: Adding the definition of how T or B cells were designated as “clonal” to the figure legend might be helpful (or at least call out in the legend that the definition is in the Methods). Presently this definition was outlined in Results, requiring the reader to have to dig a bit more to interpret the figures. Example: “T cells were designated as clonal when >2 TCRs were observed per patient (or across all patients, however you defined it).”

Some additional very minor comments are listed below, which are offered for the authors consideration. I do not view the proposed change below as essential.

Very minor comments:

- I’m a little surprised Fig. S2 is not one of (or part of) the main figures, as it’s a nice summary of the findings that is complementary to Fig. 1. I ultimately leave it to the discretion of the authors where to feature these data, and acknowledge that Fig. 1 is already quite large as is.

Response to reviewer comments

We would once again like to thank the reviewers for their helpful and constructive comments, which have helped to further strengthen our manuscript. We have now added further experimental details in the Methods section. We respond individually to the reviewers' comments below.

Reviewer 1: Comment 1

This reviewer thanks the authors for their effort in addressing the various points raised. However, such effort is limited to merely adding some points in the text, without providing any additional data or arguments to address this reviewer's previous concerns. It is actually quite telling that, in their point by point answers, the authors agree with most of such concerns.

As a consequence, my opinion has not changed. This paper in its current configuration is descriptive, does not provide the necessary information to be impactful, lacks indispensable comparators and its novelty is limited. As suggested before, this can be a useful atlas-type resource paper for a more specialised journal

Authors' response

We thank the reviewer for highlighting the limitations with our study, to which we responded in our previous revision. We hope that our results and data will be a useful resource for researchers in the future.

Reviewer 2: Comment 1

An important outstanding question in rheumatology is whether patients with RA in longstanding remission need to continue lifelong therapy or whether they can discontinue conventional synthetic disease modifying anti-rheumatic drugs (csDMARDs). Many patients who discontinue therapy eventually flare and predictors and mechanisms are unknown. Baker et al have collected an interesting cohort of 36 RA patients in remission who discontinued csDMARDs, of which, 20 flared within 6 months of follow up and 16 did not. In this study, they compared blood cell measurements of CyToF and scRNA (on a subset 8 patients who flared and 4 who did not) at baseline compared to flare or the 6 month follow up visit in those who did not flare. The analysis focused on lymphocytes and revealed changes in abundance and or gene expression in baseline versus flare. The previous submission has been effectively reorganized and focused to streamline the figures and allow bigger, more readable fonts.

The noteworthy results are:

1. Mass cytometry of samples during flare harbor increased i) CD19+CD27+CD86+CD21- B cells, ii) CD4+CD45RO+ICOS+PD1+CD38hi T cells, and iii) increased CD8+CD45RO+, PD1hiCD38hi T cells. But no differences at baseline between those who went on to flare and those who did not.

2. CITE-Seq of samples during flare harbor increased Foxp3+CD25+CTLA4+Helios+ CD4+T cells, ii) Granzyme+CD38+ CD4+T cells, iii) Proliferating CD4+T cells, iii) HLADR+CD38+CD8 T cells, iv) CXCR5+ CD8 T cells, v) IgD+CD24+ B cells, and vi) CXCR3+ B cells.

3. The most significant result is the data presented in Figure 4, which identified proteins and RNA within cell types that predicted future flare or sustained remission.

4. They also found DMARD cessation is followed by clonal expansion of CD4 and CD8 T cells.

5. They propose a model whereby Tregs of patients who are going to flare are dysfunctional and during flare, IgA plasma cells, cytotoxic CD4 and CXCR5+CD8 expand.

6. While cross sectional studies of RA flare are not difficult to collect, since patients come to clinical attention, samples from patients with impending flare are much more challenging to come by. The strength of this study is the finding in Figure 4, identified proteins and RNA within cell types that predicted future flare or sustained remission. The weakness of this data is that it is a rather small sample size and as reviewer 1 mentioned, we don't know if the patients sustained remission for longer than 6 months. The authors have addressed this in their discussion.

Authors' response

We thank the reviewer for their summary of our study. We hope that our ongoing and future work will help to further elucidate the immunological mechanisms permissive for sustained drug-free remission beyond the 6 month follow-up duration of this study.

Reviewer 3: Comment 1

This is a revised manuscript from Baker, et al., which highlights immune changes that followed removal of immunosuppressive therapies in patients who flared (compared to patients who did not flare). In my opinion, the data are of high interest to the RA community and their somewhat descriptive nature do not detract from their significance. This rich dataset should be of value to the larger rheumatology community.

Authors were generally responsive to reviewer comments and data are now presented with improved clarity. Caveats are now appropriately acknowledged, suggested citations were added/discussed, and the Discussion is improved.

Authors' response

Thank you for your comments.

Reviewer 3: Comment 2

Minor comments (revision suggested, but not required):

- *Fig. S8, S9, S10: Adding the definition of how T or B cells were designated as "clonal" to the figure legend might be helpful (or at least call out in the legend that the definition is in the Methods). Presently this definition was outlined in Results, requiring the reader to have to dig a bit more to*

interpret the figures. Example: "T cells were designated as clonal when >2 TCRs were observed per patient (or across all patients, however you defined it)."

Authors' response

Thank you for the suggestion. For Supplementary Figures 8-10, we have added the following text to the figure legends: "Clonal cells are highlighted in red, *and were defined as at least two cells that shared the same paired CDR3 nucleotide sequence within the same patient*". We have also added similar text to the legend of Figure 5: "Clonal cells were defined as at least two cells that shared the same paired CDR3 nucleotide sequence within the same patient".

Reviewer 3: Comment 3

Some additional very minor comments are listed below, which are offered for the authors consideration. I do not view the proposed change below as essential.

Very minor comments:

- I'm a little surprised Fig. S2 is not one of (or part of) the main figures, as it's a nice summary of the findings that is complementary to Fig. 1. I ultimately leave it to the discretion of the authors where to feature these data, and acknowledge that Fig. 1 is already quite large as is.*

Authors' response

We would indeed like to be able to include Supplementary Figure 2 within the main Figure 1 as per our original submission. However, the addition of further boxplots to Figure 1 in response to previous reviewer comments has reduced the available space, meaning that retaining the volcano plots in the main Figure 1 would make the text of the figures too small to read clearly. We feel that moving the volcano plots to Supplementary Figure 2 gives the best balance between preserving the clarity of the figures for the reader whilst still including the helpful volcano plot visual summary within the publication.